# The single-cell chromatin accessibility landscape in mouse perinatal testis development

**Hoi Ching Suen[1†], Shitao Rao[2,3], Alfred Chun Shui Luk[1], Ruoyu Zhang[3], Lele Yang[4], Huayu Qi[4], Hon Cheong So[3], Robin M Hobbs[5]\*, Tin-lap Lee[1]\*, Jinyue Liao[1,6]\*†**

[1]Developmental and Regenerative Biology Program, School of Biomedical Sciences, Faculty of Medicine, The Chinese University of Hong Kong, Shatin, Hong Kong, Hong Kong; [2]School of Medical Technology and Engineering, Fujian Medical University, Fujian, China; [3]Cancer Biology and Experimental Therapeutics Program, School of Biomedical Sciences, Faculty of Medicine, The Chinese University of Hong Kong, Shatin, Hong Kong, China; [4]Guangzhou Regenerative Medicine and Health Bioland Laboratory, Guangzhou Institutes of Biomedicine and Health, Guangzhou, China; [5]Germline Stem Cell Biology Laboratory, Centre for Reproductive Health, Hudson Institute of Medical Research, Melbourne, Australia; [6]Department of Chemical Pathology, The Chinese University of Hong Kong, Shatin, New Territories, Hong Kong, China

**\*For correspondence:**
robin.hobbs@monash.edu (RMH);
tinlaplee@gmail.com (T-IL);
Liaojinyue@cuhk.edu.hk (JL)

†These authors contributed equally to this work

**Competing interest:** The authors declare that no competing interests exist.

**Abstract** Spermatogenesis depends on an orchestrated series of developing events in germ cells and full maturation of the somatic microenvironment. To date, the majority of efforts to study cellular heterogeneity in testis has been focused on single-cell gene expression rather than the chromatin landscape shaping gene expression. To advance our understanding of the regulatory programs underlying testicular cell types, we analyzed single-cell chromatin accessibility profiles in more than 25,000 cells from mouse developing testis. We showed that single-cell sequencing assay for transposase-accessible chromatin (scATAC-Seq) allowed us to deconvolve distinct cell populations and identify *cis*-regulatory elements (CREs) underlying cell-type specification. We identified sets of transcription factors associated with cell type-specific accessibility, revealing novel regulators of cell fate specification and maintenance. Pseudotime reconstruction revealed detailed regulatory dynamics coordinating the sequential developmental progressions of germ cells and somatic cells. This high-resolution dataset also unveiled previously unreported subpopulations within both the Sertoli and Leydig cell groups. Further, we defined candidate target cell types and genes of several genome-wide association study (GWAS) signals, including those associated with testosterone levels and coronary artery disease. Collectively, our data provide a blueprint of the 'regulon' of the mouse male germline and supporting somatic cells.

## Editor's evaluation

This manuscript by Liao et al. aims to understand the genetic networks that underlie or modulate gonadogenesis and germ cell maturation during the fetal to neonatal transition. This goal was achieved by performing scATACseq on multiple timepoints (E18.5 and Postnatal days 1,2,5). Clustering of thousands of cells has striking cellular diversity and convincingly led to the identification of additional novel populations, of both germ cell and somatic origins. This is an important paper with far-reaching implications in reproductive biology, but additional validation would be needed to confirm the correlative observations and the functionality of newly identified testis cells.

## Introduction

Mammalian testis consists of germ cells and distinct somatic cell types that coordinately underpin the maintenance of spermatogenesis and fertility. These testicular cells display extensive developmental dynamics during the perinatal period. Primordial germ cells give rise to M-prospermatogonia at about embryonic day (E) 12, which enter G0 mitotic arrest at about E14 to form T1-prospermatogonia (T1-ProSG) (*McCarrey, 2013*). Shortly after birth, T1-ProSG resume mitotic activity and begin migrating from the center of the testis cords to the basal lamina of testicular cords, and become T2-prospermatogonia (T2-ProSG). Once resident at the basement membrane, T2-ProSG generate spermatogonia including self-renewing spermatogonial stem cells (SSCs) or directly transition into differentiating spermatogonia that participate in the first round of spermatogenesis (*Kluin and de Rooij, 1981*; *Manku and Culty, 2015*).

Specialized somatic cells play a pivotal role in maintaining normal germ cell development and spermatogenesis. SSCs and their initial progenies reside in a niche on the basement membrane and are surrounded by Sertoli cells, which nourish SSCs. In mice, Sertoli cells actively proliferate during the neonatal period for 2 weeks (*Vergouwen et al., 1991*). Outside the seminiferous tubules, the interstitial compartment of testis mainly contains stroma, peritubular myoid cells (PTMs), Leydig cells, macrophages, and vascular cells. PTMs are smooth muscle cells that distribute over the peripheral surface of the basement membrane (*Maekawa et al., 1996*). They are mainly involved in tubule contractions to facilitate the movement of sperm to the epididymis and secrete extracellular matrix materials (*Chen et al., 2014*). Leydig cells are responsible for steroidogenesis and provide critical support for spermatogenesis. In mammals, there are two types of Leydig cells, fetal Leydig cells (FLCs) and adult Leydig cells (ALCs), which develop sequentially in the testis (*Shima, 2019*). While FLCs start to degenerate after birth, they are replaced by stem Leydig cells (SLCs), which are the progenitors of ALCs (*Su et al., 2018*).

One of the goals of developmental biology is to identify transcriptional networks that regulate cell differentiation. Recent advances in single-cell transcriptomic methods have enabled an unbiased identification of cell types and gene expression networks in testis (*Green et al., 2018*; *Tan et al., 2020*). A key remaining question is how the gene expression is dynamically regulated during the development of distinct cell types. Since all cells share the same genetic information, cell-type development must be regulated by differential chromatin accessibility in a cell type-specific manner. Thus, the understanding of the testis development, especially the cell type-specific transcription factors (TFs), is of paramount importance. scRNA-Seq provides limited information of TFs, which are usually lowly expressed. Although sequencing methods such as ATAC-Seq and DNase-Seq have been developed for profiling chromatin accessibility landscapes across samples and classification of regulatory elements in the genome, the nature of bulk measurements masks the cellular and regulatory heterogeneity in subpopulations within a given cell type (*Shema et al., 2019*). Moreover, uncovering regulatory elements in testicular cell types has been particularly challenging as samples are limited and heterogeneous. Notably, only one previous study examined the genome-wide chromatin accessibility in Sertoli cells, using the *Sox9* transgenic line followed by DNase-Seq (*Maatouk et al., 2017*).

By profiling the genome-wide regulatory landscapes at a single-cell level, recent single-cell sequencing assay for transposase-accessible chromatin (scATAC-Seq) studies have demonstrated the potential to discover complex cell populations, link regulatory elements to their target genes, and map regulatory dynamics during complex cellular differentiation processes. We reasoned that the advent of new single-cell chromatin accessibility sequencing methods, combined with single-cell transcriptomic data, would be instrumental in advancing our understanding of gene regulatory networks in mammalian testis development. Here, we applied scATAC-Seq to deconvolve cell populations and identify cell type-specific epigenetic regulatory circuits during perinatal testis development. The dataset led to identification of key cell type-specific TFs, defined the cellular differentiation trajectory, and characterized regulatory dynamics of distinct cell types. Furthermore, our results shed light on the identification of target cell types for genetic variants. To enable public access to our data, we constructed the mouse testis epigenetic regulatory atlas website at http://testisatlas.s3-website-us-west-2.amazonaws.com/.

## Results

### Single-cell ATAC-Seq captures developmental and cell type-specific heterogeneity in the testis

To delineate the dynamic changes on cellular populations in a developing testis, we profiled the chromatin accessibility landscapes of mouse perinatal testis across E18.5 and postnatal stages (P0, P2, and P5) by scATAC-Seq (*Figure 1A*). These time points were chosen to represent the diversity of cell-type compositions involved in the key developmental events in the testis (*Figure 1B*). Altogether, we profiled chromatin accessibility in 25,613 individual cells after stringent quality control filtration and heterotypic doublet removal (*Figure 1—figure supplement 1*). These samples showed no clustering based on covariates such as transcription start site (TSS) enrichment and number of fragments detected (*Figure 1—figure supplement 2A*).

Several clusters showed developmental stage specificity, which were made up almost entirely of cells from a single time point (*Figure 1—figure supplement 2B*). To improve cell-type annotation, we used Harmony to integrate datasets of four time points and project cells onto a shared embedding in which cells were grouped by cell type rather than developmental stage (*Korsunsky et al., 2019*). Unbiased iterative clustering of these single cells after integration identified 11 distinct clusters (*Figure 1—figure supplement 2B*). Some clusters could be assigned to known testicular cell types based on gene activity scores of key marker genes compiled from chromatin accessibility signals within the gene body and promoter (*Figure 1C*; *Tan et al., 2020*).

While this approach provided broad classifications for cell-type annotation, an unbiased method is needed for more accurate classification. Therefore, we leveraged a previously published scRNA-Seq dataset of perinatal testis samples to predict cell types in scATAC-Seq data (*Tan et al., 2020*). We first re-analyzed scRNA-Seq data to determine the cellular composition and annotate cells based on their transcriptional profiles. Prediction of cell types in scATAC-Seq was then performed by directly aligning cells from scATAC-Seq with cells from scRNA-Seq through comparing the 'query' gene activity scores matrix with 'reference' gene expression matrix based on the top variable genes in the scRNA-Seq dataset (*Supplementary file 1*). The results showed that the vast majority of cells had a high prediction score and were confidently assigned to a single cell type (*Figure 1D*, *Figure 1—figure supplement 2C*). Cell-type proportions varied across time points, such as the expansion of germ cells during the early neonatal period (*Figure 1E*). We further validated the cluster assignment by gene score and chromatin accessibility profiles of marker genes (*Figure 1F*). Taken together, scATAC-Seq allowed the detection and assignment of cell identities in the developing testis.

### Chromatin accessibility defines cell types in developing testis

Cell types can be distinguished based on whether differentially accessible chromatin regions (DARs) are 'open' or 'closed'. After identifying 214,890 accessible chromatin regions in the scATAC-Seq library (*Supplementary file 2*), we investigated cell type-specific chromatin accessibility profiles. We compared differences in chromatin accessibility among cell types directly using Wilcoxon testing to identify DARs while accounting for TSS enrichment and the number of unique fragments per cell (*Figure 2A*, *Supplementary files 3 and 4*). Deconvolution of chromatin accessibility by cell types revealed that accessible sites are primarily located in the distal and intron region (>3 kb from TSS), suggesting an enrichment of gene regulatory elements (*Figure 2—figure supplement 1A*).

We found cluster-specific DARs were associated with cell type-specific marker genes identified from scRNA-Seq (*Figure 2B*). For example, *Amh* is a marker gene in Sertoli cells, and it showed increases in both number and amplitude of ATAC peaks within its promoter and gene body. We further compared DARs to a previously published DNase-Seq experiments in bulk Sertoli cells and found that DNase I hypersensitive sites were clearly enriched in the Sertoli cell population in our scATAC-Seq (*Figure 2—figure supplement 1B*; *Maatouk et al., 2017*). These data confirm that scATAC-Seq is a robust method for the detection of cell type-specific chromatin accessibility.

### Chromatin accessibility is associated with cell type-specific TF activity

Currently, the identities of cell type-specific TFs involved in testis development are poorly defined. Accessibility at regulatory sites is driven by TF binding and histone modifications of local chromatin (*Cui et al., 2013*). To characterize the determinants of chromatin accessibility variation among cell

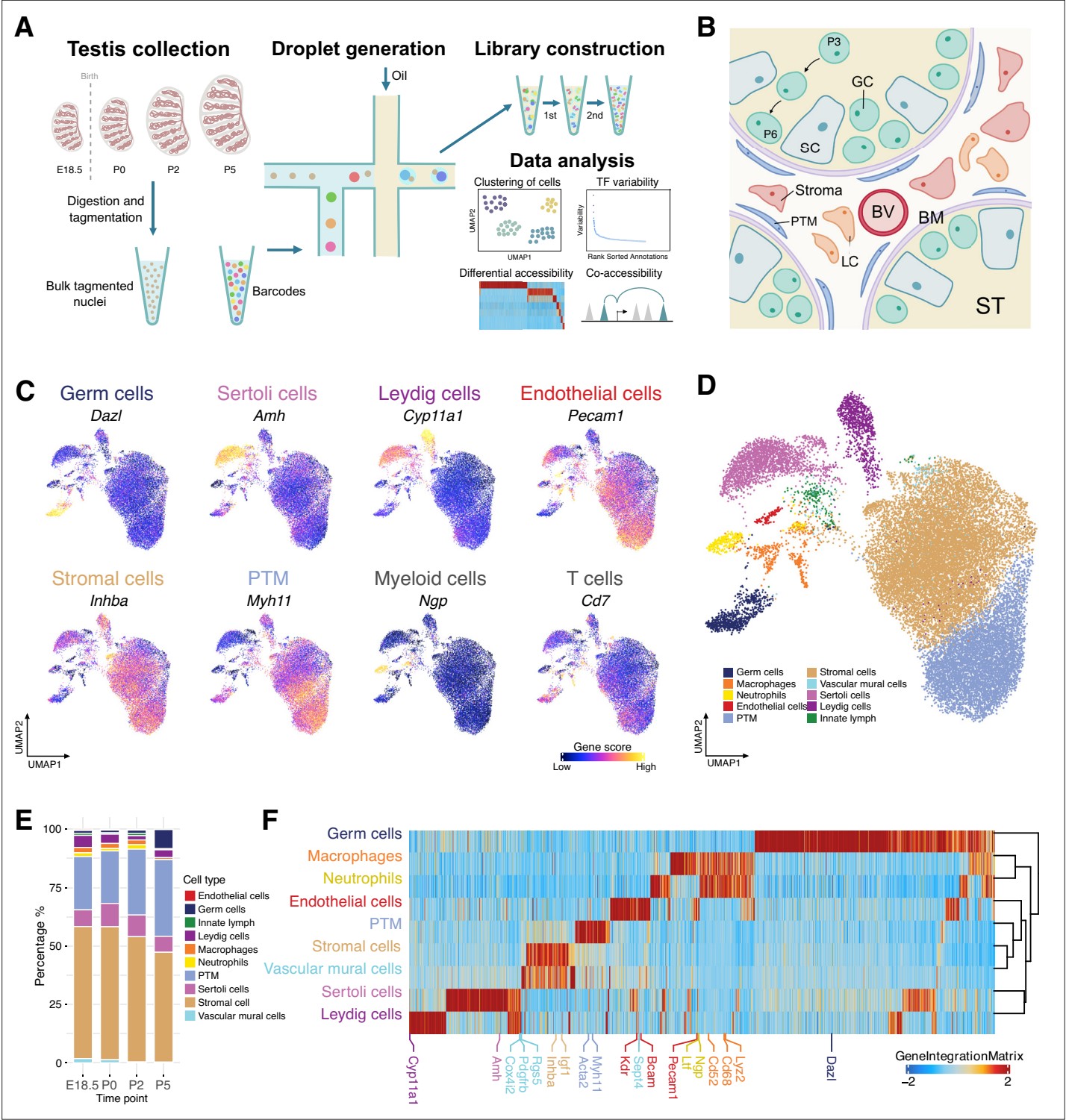

**Figure 1.** Classification and identification of germ cells and somatic cells during perinatal testicular development. (**A**) Experimental design. The workflow of testis collection and single-cell sequencing assay for transposase-accessible chromatin (scATAC-Seq) to measure single nuclei accessibility on BioRad SureCell ATAC-Seq platform. (**B**) Illustration of the testicular microenvironment. GC: germ cell; SC: Sertoli cell; LC: Leydig cell; BV: blood vessel; BM: basement membrane; ST: seminiferous tubule; PTM: peritubular myoid cell. (**C**) Uniform manifold approximation and projection (UMAP) representations with cells colored by the gene score of marker genes for each cell type. (**D**) UMAP representation of cells captured from four time points. Cells are colored by predicted groups. (**E**) Bar chart showing the distribution of cells in each cluster for different time points. (**F**) Heatmap of 12,250 marker genes across cell types (FDR ≤ 0.05, Log2FC ≥ 0.2).

*Figure 1 continued on next page*

*Figure 1 continued*

The online version of this article includes the following figure supplement(s) for figure 1:

**Figure supplement 1.** Quality assessment metrics for single-cell sequencing assay for transposase-accessible chromatin (scATAC-Seq) libraries, related to *Figure 1*.

**Figure supplement 2.** Testicular cell-type annotation and distribution, related to *Figure 1*.

types, we predicted TF 'activity' for individual cell types based on the presence of binding motifs within DARs. Assessment of enriched TFs and their cognate motifs identified several known cell type-specific regulators – including the nuclear receptors (NR4A1 and NR5A1) in Sertoli cells and Leydig cells, ESR2 in Leydig cells, MYOG in PTMs, and previously uncharacterized TFs as potential cell type-specific regulators (*Figure 2C*). For example, we found that ZEB1 and SNAI2 motifs were enriched in germ cells, indicating they may undergo mesenchymal-like transition in perinatal development (*Hammoud et al., 2015*; *Liao et al., 2020*). DNA bound by TFs is protected from transposition by Tn5, which can be visualized by plotting the 'footprint' pattern of each TF as the local chromatin accessibility surrounding the motif midpoint. Examining the footprint validated the cell type-dependent differential footprint occupancy of identified TFs (*Figure 2D*).

Although motif enrichment for DAR can be informative, this measurement is not calculated on a per-cell basis and they do not take into account the insertion sequence bias of Tn5. Therefore, in the second analysis approach, we used chromVAR to infer TF motif activity, which can reflect the enrichment level of the TF motif in accessible regulatory elements in an scATAC-Seq dataset. We first identified deviant TF motifs by stratifying motifs based on the degree of variation observed across clusters. Since TFs from the same family often share a similar motif, this makes it challenging to identify the specific TFs that actually drive the observed changes in chromatin accessibility. This is illustrated by SOX9, which exhibited enriched TF activity in both Sertoli and germ cell clusters, despite being known to be expressed only in Sertoli cells. This could be attributed to the expression of other SOX family genes, such as SOX3, which share similar DNA binding motifs, in germ cells (*Raverot et al., 2005*). However, SOX9 exhibited concordant enrichment of TF activity and gene expression only in Sertoli cells and can be considered a strong candidate in Sertoli cells, but not in germ cells.

To reduce false discovery, we systematically identified putative positive regulators determined from the correlation between the gene expression based on scRNA-Seq dataset (or inferred gene activity based on scATAC-Seq dataset) and the chromVAR motif activity score, reasoning that expression of high-confidence TFs is correlated with their motif accessibility (*Figure 2E*). Clustering analysis of positive regulators showed that diverse combinatorial TF motif landscapes were apparent across cell types and closely mirrored gene accessibility profiles of respective TFs (*Figure 2F–H*). There was an increased GATA1 TF 'activity' (motif activity) in the Sertoli cell cluster, in addition to increased chromatin accessibility in *Gata1* (gene activity) and increased *Gata1* transcription (gene expression) (*Figure 2G*). It has been shown that mutation of GATA1 causes human cryptorchidism (*Nichols et al., 2000*) and its expression in Sertoli cells is conserved between human and mouse (*Yomogida et al., 1994*). A similar pattern was seen for DMRT1 in germ cells, which is one of the top motifs enriched in human SSC-specific ATAC-Seq peaks (*Guo et al., 2017*). This analysis also revealed shared and unique regulatory programs across cell types. For example, PTM and stromal cells shared similar regulators, but PTM demonstrated higher activity of AR and TCF21 (*Figure 2G*). Similarly, NR5A1 and GATA4 were more active in both Leydig cell and Sertoli cell populations. However, only Sertoli cells showed increased activity in the SOX family.

Importantly, we observed that individual cell types can be defined by TF 'activity', suggesting that cell type-specific TFs likely regulate chromatin accessibility. Collectively, these results are indicative of robust inference of TF activity at the level of single cells and reveal TF dynamics central to *cis*-regulatory specification of diverse cell states.

## Chromatin accessibility is associated with cell type-specific chromatin interaction networks

As enhancers play a critical role in establishing tissue-specific gene expression patterns during development, we predicted that active enhancers would be enriched around lineage-specific genes. To test this, we used an analytical framework to link distal peaks to genes in *cis*, based on the coordination

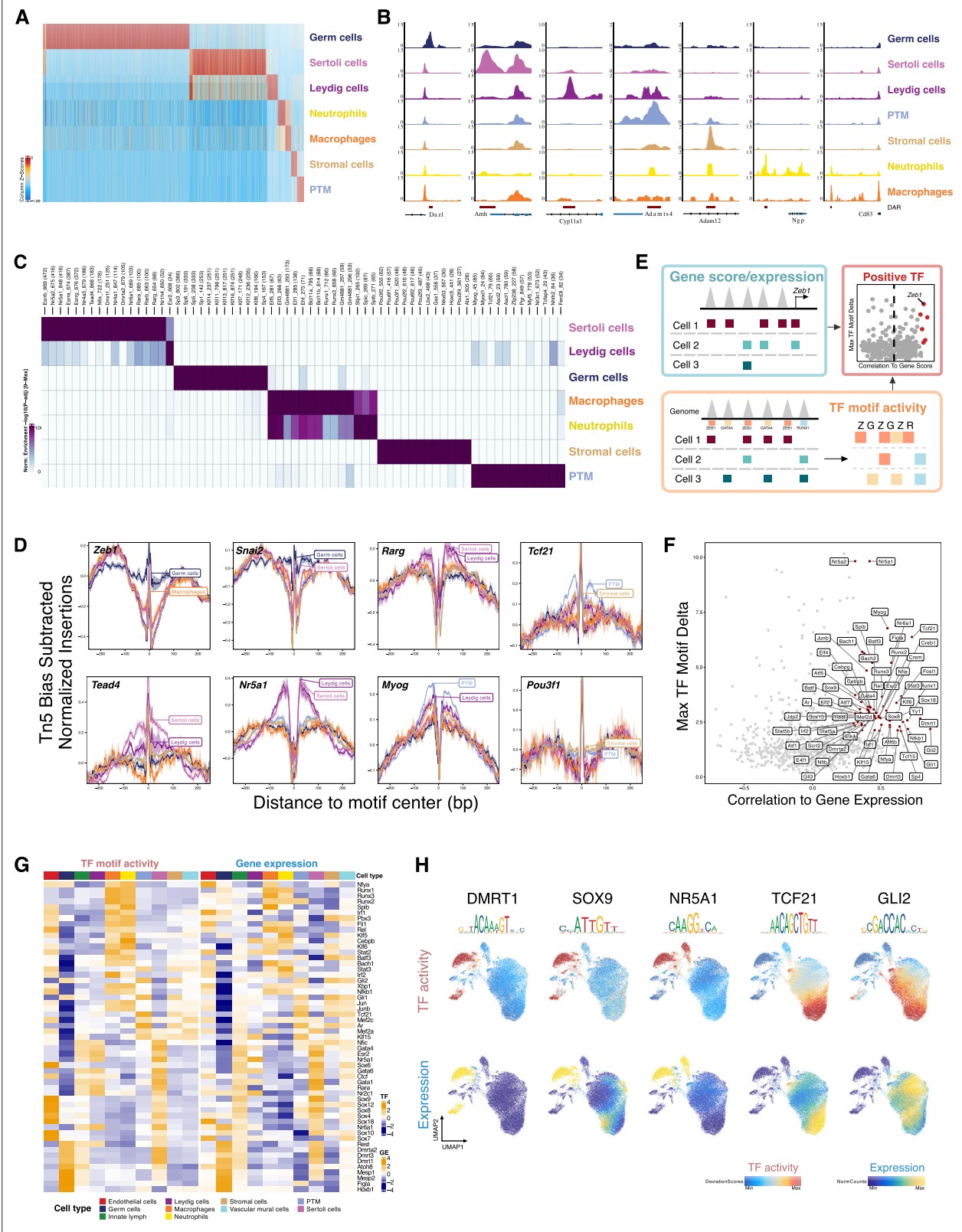

**Figure 2.** Characterization of differentially accessible regions and identification of cell type-specific transcription factors (TFs). (**A**) Heatmap of 51,937 differentially upregulated accessible peaks (FDR ≤ 0.01, Log2FC ≥ 2) across cell types. (**B**) Aggregated single-cell sequencing assay for transposase-accessible chromatin (scATAC-Seq) profiles of selected markers. (**C**) Heatmap of enriched motifs (FDR ≤ 0.1, Log2FC ≥ 0.5) across cell types. (**D**) TF footprints (average ATAC-Seq signal around predicted binding sites) for selected TFs. (**E**) Schematic of identifying positive TF regulators through

*Figure 2 continued*

correlating gene score (scATAC-Seq data)/gene expression (integrating scATAC-Seq and scRNA-Seq data) with TF motif activity (scATAC-Seq data). (**F**) Scatter plot of positive TF regulators (correlation >0.5, adjusted p-value <0.01). (**G**) Heatmaps of differential TF motif activity (left) and gene expression (right) of positive TF regulators in F. (**H**) TF overlay on scATAC uniform manifold approximation and projection (UMAP) of TF chromVAR deviations (top) and gene expression (bottom).

The online version of this article includes the following figure supplement(s) for figure 2:

**Figure supplement 1.** Characterization of cell type-specific differentially accessible chromatin regions (DARs), related to *Figure 2*.

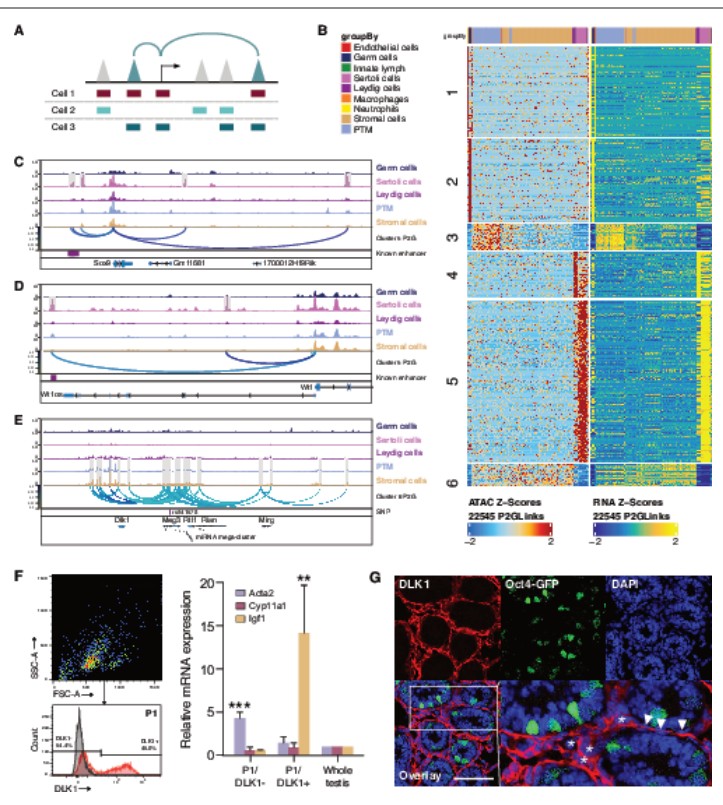

**Figure 3.** Chromatin interaction networks in different cell types. (**A**) Schematic for identifying significant peak-to-gene links by correlating accessible peaks (single-cell sequencing assay for transposase-accessible chromatin [scATAC-Seq] data) to gene expression (integrating scATAC-Seq data and scRNA-Seq data). (**B**) Heatmaps of peak accessibility (left) and gene expression (right) of 22,545 peak-to-gene linkages across cell types. (**C**) Aggregated scATAC-Seq profiles showing peak-to-gene links to the *Sox9* locus overlapped with known enhancer regions. (**D**) Aggregated scATAC-Seq profiles showing peak-to-gene links to the *Wt1* locus overlapped with known enhancer regions. (**E**) Aggregated scATAC-Seq profiles showing peak-to-gene links to the *Dlk1* locus overlapped with SNP. (**F**) Sorting strategy for isolation of DLK1- and DLK1+ cells from P6 whole testis. The majority of DLK1+ cells are located in P1 (upper left). The DLK1-/+ population was gated using Red-X-labeled sample compared with unstained control (lower left). RT-PCR analysis (right) of relative expression of peritubular myoid cell (PTM) marker (*Acta2*), Leydig cell marker (*Cyp11a1*), and stromal cell marker (*Igf1*) of DLK1-/+ cells compared with whole testis sample (p<0.001, n=3, one-way ANOVA). *Gapdh* was used as endogenous control. Error bars are plotted with SD. (**G**) Representative confocal images of testis sections from Oct4-GFP transgenic mice at P6. Stromal cells (asterisks) and some PTMs (arrowheads) are positive for DLK1 (red). Oct4-GFP indicated germ cells. Cell nuclei were stained with DAPI. Scale bar = 50 µm.

The online version of this article includes the following figure supplement(s) for figure 3:

**Figure supplement 1.** Characterization of peak-to-gene links across cell types, related to *Figure 3*.

**Figure supplement 2.** Aggregated single-cell sequencing assay for transposase-accessible chromatin (scATAC-Seq) profiles showing peak-to-gene links to the selected gene examples, related to *Figure 3*.

of chromatin accessibility and gene expression levels across cells (*Figure 3A*). We identified 35,245 peak-to-gene links by correlating accessibility changes of ATAC peaks within 250 kb of the gene promoter with the mRNA expression of the gene from scRNA-Seq. Some of these peak-to-gene links are likely to be promoter-enhancer regulatory units, as 3262 regions overlapped with previously identified testis enhancers (*Gao and Qian, 2020*; *Figure 3—figure supplement 1A*).

To identify putative *cis*-regulatory elements (CREs) specific to each cell type, we performed clustering analysis, with each cluster containing peak-to-gene links enriched in one or two specific cell types (*Figure 3B*). Gene Ontology (GO) analysis of the targets of peak-to-gene links in each cluster confirmed that they were highly enriched in terms related to regulations of each cell type (*Figure 3—figure supplement 1B*). The full list of peak-to-gene links in each cluster can be found in *Supplementary file 5*.

We next examined whether we can use this information to link DARs to known cell type-specific enhancers. During male sex determination, *Sry* activates male-specific transcription of *Sox9* in the male genital ridge via the testis-specific enhancer core element (TESCO) enhancer (*Sekido and Lovell-Badge, 2008*). Comparison of the genomic region around *Sox9* among all cell types revealed a region 13 kb upstream formed a peak-to-gene link with the *Sox9* TSS, which is unique to the Sertoli cell population and overlapped with the 3 kb TES enhancer including the TESCO elements (*Figure 3C*). Enhancer activity was previously narrowed to a subregion of TES: the 1.3 kb TESCO element (*Sekido and Lovell-Badge, 2008*). Besides known enhancers, our data linked an additional three elements located 9 kb 5′, 21 kb 3′, and 68 kb 3′ to *Sox9* representing novel regulatory elements of *Sox9* gene regulation. Our analysis also successfully revealed a previously identified functional enhancer as a novel candidate to regulate Sertoli cell marker *Wt1* (*Figure 3D*). After confirming that our approach can be used to reveal functionally relevant regulatory regions, we next identified putative regulatory elements for important cell type-specific regulators in each cell type, including *Nanos2* and *Uchl1* in germ cells, *Cyp11a1* and *Nr5a1* in Leydig cells, *Tpm1* and *Socs3* in PTMs (*Figure 3—figure supplement 2*), and *Dlk1* in stromal cells and PTMs (*Figure 3E*).

Although DLK1 is considered as a marker for immature Leydig cells in human (*Lottrup et al., 2014*), the *Dlk1-Gtl2* locus demonstrated preferential accessibility in stromal cells and PTMs. This coincided with the high number of peak-to-gene links including the largest mammalian miRNA mega-cluster located approximately 150 kb downstream (*Seitz et al., 2004*). Therefore, we examined the expression of *Dlk1* in mouse testicular cells. We sorted DLK1+ mouse testicular cells using fluorescence-activated cell sorting and observed higher expression of *Igf1* mRNA compared to whole testis samples. This suggests that the sorted cells were enriched for stromal cells, since *Igf1* is commonly used as a marker for this cell type (*Figure 3F*). Immunostaining of neonatal testis tissue demonstrated that stromal cells and PTMs were positive for DLK1 (*Figure 3G*).

In conclusion, our results highlight the occurrence of diverse cell type-specific regulatory configurations among CREs and their target genes in the testis.

## Stage-specific TF regulators and chromatin co-accessibility during gonocyte to spermatogonia transition

Next, we analyzed the chromatin accessibility characteristics of the individual germ cell subsets in our datasets. Since the goal was to reveal developmental dynamics, we did not perform 'harmony' integration since we didn't want to remove the contribution of developmental stage-of-origin from the embedding. Re-clustering of germ cells from the E18.5, P0, P2, and P5 testicular datasets revealed seven-cell clusters (*Figure 4A* and *Figure 4—figure supplement 1A*). Notably, germ cells from E18.5 and P0 are largely clustered together and occupy clusters GC1 and GC3, indicating a minimal change of chromatin accessibility before and after birth. In contrast, P2 cells are present in GC2 and P5 cells occupy the remaining clusters.

Deconvoluting cell states using scATAC-Seq measurements alone is difficult within a single cell type. Therefore, we integrated the germ cell subsets based on scATAC-Seq data with the published perinatal testis scRNA-Seq dataset (*Figure 4—figure supplement 1B*). The prediction scores of individual cells were overall high, indicating the cluster identity assignment was reliable (*Figure 4—figure supplement 1C*). This prediction revealed four clusters of known developmental stages, T1-ProSG (T1), T2-ProSG (T2), undifferentiated spermatogonia (Undiff), and differentiation-primed spermatogonia (Diff), together with two clusters with unknown identity (*Figure 4A*).

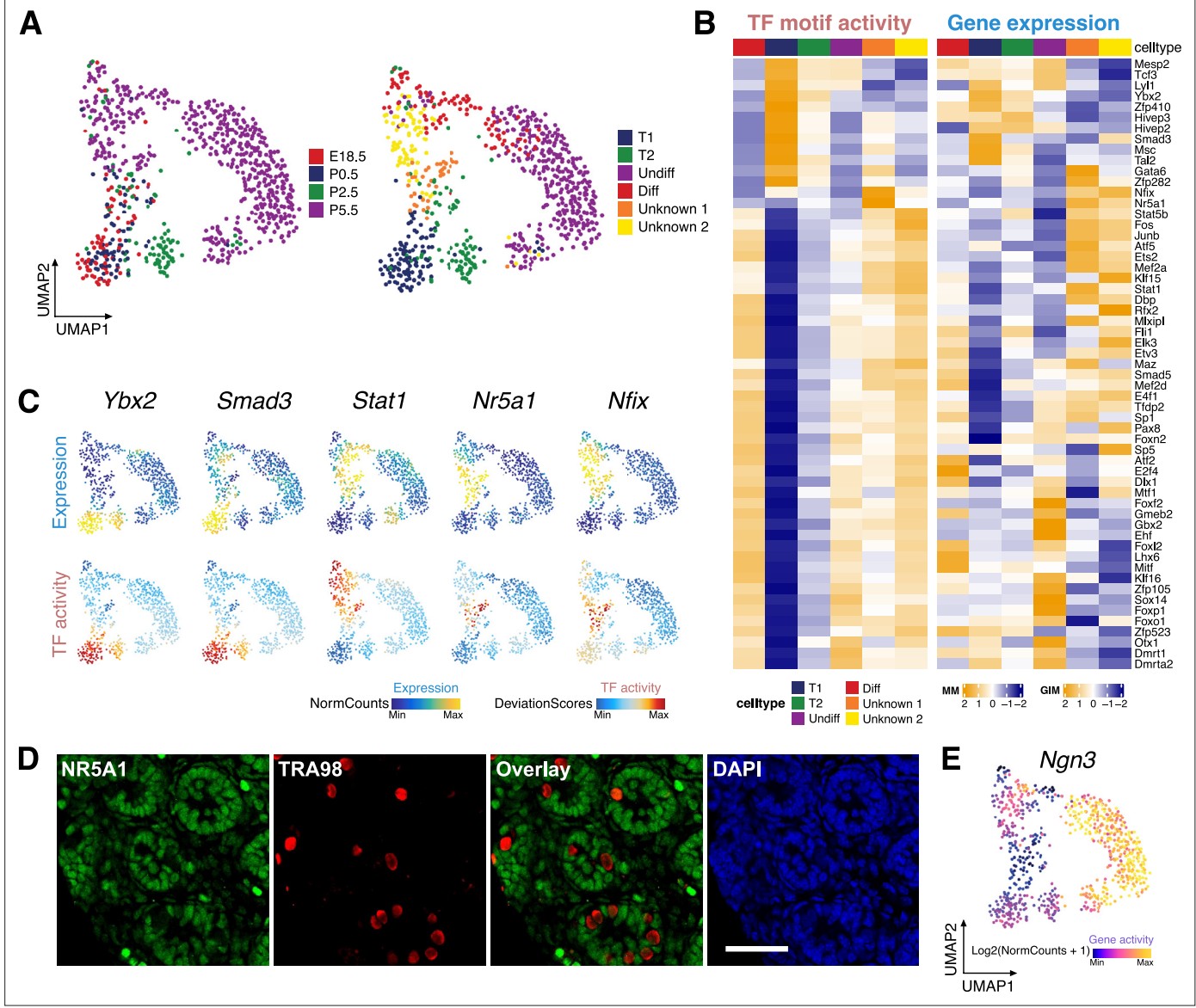

**Figure 4.** Identification of germ cell clusters during the perinatal period. (**A**) Uniform manifold approximation and projection (UMAP) representation of germ cells. Cells are colored by time points (left) and clustering based on constrained integration with scRNA-Seq data (right). (**B**) Heatmaps of differential transcription factor (TF) motif activity (left) and gene activity (right) of positive TF regulators across cell clusters (correlation >0.5, adjusted p-value <0.01). (**C**) TF overlay on scATAC UMAP of gene expression (top) and TF chromVAR deviations (bottom) for positive TF regulator examples in B. (**D**) Representative confocal images of immunostaining on sections from P6 testis demonstrate that a subset of germ cells (TRA98+) express the Sertoli cell marker NR5A1 (arrowhead), while the majority of germ cells are NR5A1-negative (arrow). Scale bar = 50 μm. (**E**) Gene activity of *Ngn3* shown in UMAP.

The online version of this article includes the following figure supplement(s) for figure 4:

**Figure supplement 1.** Identification of germ cells during the perinatal period, related to *Figure 4*.

**Figure supplement 2.** Reconstruction of germ cell developmental trajectory, related to *Figure 4*.

**Figure supplement 3.** Heatmaps showing peak accessibility (left) and gene expression (right) of 12,819 peak-to-gene linkages across germ cell clusters, related to *Figure 4*.

**Figure supplement 4.** Aggregated single-cell sequencing assay for transposase-accessible chromatin (scATAC-Seq) profiles showing germ cell peak-to-gene links to the selected gene examples from *Figure 4—figure supplement 3*, related to *Figure 4*.

We first identified the TFs important for each cluster, revealing 57 putative positive TF regulators in germ cell development (*Figure 4B and C*). For instance, consistent with previous studies, *Foxo1* and *Dmrt1* exhibited increased TF motif activity and gene expression in undifferentiated spermatogonia (*Goertz et al., 2011*; *Matson et al., 2010*). *E2f4* shows highest activity in differentiation-primed spermatogonia, and is known to be critical for the development of the male reproductive system (*Danielian et al., 2016*). Interestingly, this analysis identified *Nr5a1* as the positive TF regulators in the unknown clusters (*Figure 4B and C*, *Figure 4—figure supplement 1D*). Since *Nr5a1* is widely considered as a somatic cell marker in testis (*Luo et al., 1994*), we performed immunostaining to examine NR5A1's expression in germ cells (*Figure 4D*). Indeed, a subset of germ cells expressed NR5A1, which ruled out the possible contamination from somatic cells. Interestingly, previous scRNA-Seq of neonatal pro-spermatogonia identified a spermatogonial signature cluster showed high levels of mRNAs characteristic of Sertoli cells, including *Nr5a1*, *Sox9*, and *Wt1* (*Hermann et al., 2015*). Additionally, the expression of the somatic cell marker WT1 has been observed in some germ cells through immunostaining (*Wen et al., 2021*). These observations reinforced our findings that cells with germ cell identity can express somatic cell genes (*Figure 4—figure supplement 1D*). Our results also revealed several TF candidates regulating T1-ProSG, which have previously been difficult to identify due to technical challenges in isolating this cell population, such as *Ybx2*, *Smad3*, and *Msc*. In line with its role in testicular development, *Gata6* is upregulated in T1- and T2-ProSG (*Padua et al., 2015*).

We then aimed to reconstruct the differentiation trajectories by ordering the clusters with developmental stages predicted by scRNA-Seq integration. Two possible trajectories were observed, as germ cell differentiation appeared to diverge at P0 via two distinct branches. The first trajectory represents the differentiation fit into the conventional model as it charts a trajectory from gonocyte to undifferentiated and then differentiating spermatogonia in P5. The second path bypassed the undifferentiated state but passed through the unknown populations and directly reached the differentiating state by P5. It has been reported that the first round of spermatogonia arise from a unique neurogenin-3 (*Ngn3*) negative pool of ProSG that transitions directly into A1 spermatogonia (*Yoshida et al., 2006*). Interestingly, the unknown population (Unknown-2) displayed the lowest level of *Ngn3* gene activity, which raised the possibility that the second trajectory represents the origin of the first wave of spermatogenesis (*Figure 4E*).

To determine the key genes driving the spermatogonial development in the first trajectory, we generated a pseudotime trajectory and uncovered a list of genes with dynamic changes (*Figure 4—figure supplement 1E and F*). The pattern of TF dynamics suggested a model of differentiation as a transition between two phases involving progressive loss of gonocyte-specific TF activities and gradual increase of TF activities relevant to spermatogonia. We observed that the motif binding activity of *Id4* was initially high but then declined after birth, while that of ETS and Sp/KLF family members increased in spermatogonia (*Figure 4—figure supplement 1E*). To further reveal TFs that drive the germ cell development, we pruned the data by correlating gene expression of a TF to its corresponding TF z-score (*Figure 4—figure supplement 2A and B*). This method accurately identified recognized regulators in spermatogonial differentiation. For example, *Sohlh2*, which is critical for early spermatogenesis, is more accessible at the late stage (*Hao et al., 2008*). This also raised some novel candidates regulating spermatogonial differentiation, such as *Ybx2* and *Atf4*, which are both essential to male fertility (*Fischer et al., 2004*; *Yang et al., 2005*).

We further predicted regulatory interactions from scATAC-Seq data, identifying 12,819 putative peak-to-gene links (*Figure 4—figure supplement 3*). For example, Cluster 1 included peaks linked to stem cell-related genes *Sdc4* and *Cdh1* (*Figure 4—figure supplement 4*). Cluster 2 included peaks connected to genes related to progenitors and differentiating cells such as proliferation marker *Top2a*. Cluster 3 peak-to-gene links were more accessible predominantly in T1-ProSG, such as *T* and *Fbxo4*. Taken together, our results could lead to a deeper understanding of the expression pattern of TF regulators in cells of the developing testis and also reveal candidate CREs essential for regulating spermatogonial development and differentiation.

## TF dynamics during perinatal Sertoli cell development

Re-clustering of Sertoli cells revealed six-cell clusters (*Figure 5A*). SC1 is the most varied from the other clusters, with marker genes related to spermatogenesis, such as *Fzr1*, *Egfr*, and *Npas2* (*Figure 5—figure supplement 1A*). We then identified TFs enriched along the Sertoli cell developmental

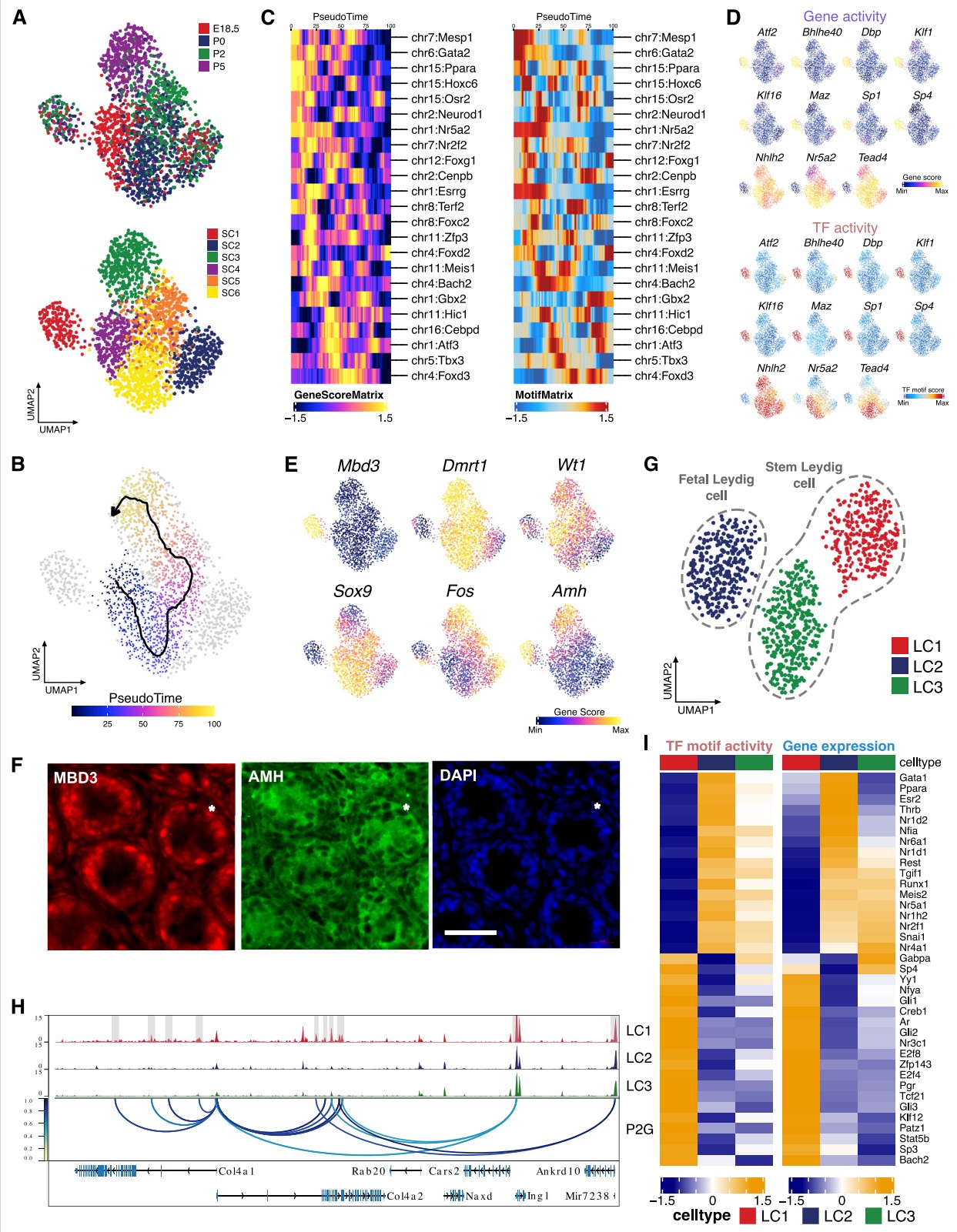

**Figure 5.** Identification of Sertoli and Leydig cell clusters during the perinatal period. (**A**) Uniform manifold approximation and projection (UMAP) representation of Sertoli cells. Cells are colored by time points (upper panel) and cell clusters (lower panel). (**B**) Single-cell sequencing assay for transposase-accessible chromatin (scATAC-Seq) profiles are ordered by pseudotime, corresponding to the perinatal development trajectory. (**C**) Smoothened heatmaps showing dynamic gene score (left) and motif accessibility (right) of indicated transcription factors (TFs) along pseudotime for

*Figure 5 continued on next page*

*Figure 5 continued*

gene-motif pairs of the trajectory in B. (**D**) TF overlay on scATAC UMAP of gene activity scores (top) and TF chromVAR deviations (bottom) for positive TF regulators in C. (**E**) Gene activity scores of *Mbd3*, *Fos*, and Sertoli cell marker genes (*Dmrt1*, *Wt1*, *Amh*, and *Sox9*) shown in UMAP. (**F**) Representative confocal images of immunostaining on sections from P6 testis reveal that Sertoli cells exhibit diverse expression patterns of MBD3 and AMH, such as MBD3-high/AMH-high (arrowhead), MBD3-low/AMH-low (arrow), and MBD3-low/AMH-high (asterisk). Cell nuclei were stained with DAPI. Scale bar = 50 µm. (**G**) UMAP representation of Leydig cells. Cells are colored by cell clusters. (**H**) Aggregated scATAC-Seq profiles showing peak-to-gene links to the *Col4a1* and *Col4a2* loci in C1 cluster. (**I**) Heatmaps of differential TF motif activity (left) and gene expression (right) of positive TF regulators (correlation >0.5, adjusted p-value <0.01).

The online version of this article includes the following figure supplement(s) for figure 5:

**Figure supplement 1.** Identification of Sertoli and Leydig cell clusters during the perinatal period, related to *Figure 5*.

**Figure supplement 2.** Identification of peritubular myoid cell (PTM) and stromal cell clusters during the perinatal period.

**Figure supplement 3.** Identification of blood cell clusters during the perinatal period.

trajectory (*Figure 5B and C*). For instance, *Gata2* and *Ppara* were enriched at early developmental stages (*Figure 5—figure supplement 1B*). *Gata2* has been identified as a target of androgen receptor (AR) in Sertoli cells, while *Ppara* regulates cholesterol metabolism and lipid oxidation in Sertoli cells (*Bhardwaj et al., 2008*; *Shi et al., 2018*). In contrast, *Hic1* and *Cebpd* were upregulated at the later stage (*Figure 5—figure supplement 1B*). Conditional knockout of *Hic1* in mice resulted in fewer Sertoli cells in seminiferous tubules (*Uchida et al., 2020*). Further, induction of C/EBP proteins by cAMP may play a role in FSH-dependent regulation in Sertoli cells (*Grønning et al., 1999*).

We further uncovered the positive TF regulators in distinct Sertoli cell subsets. Interestingly, the SC1 cluster could be clearly distinguished from the other cell clusters as the 11 identified potential positive TF regulators either have highest or lowest activity in SC1 (*Figure 5D*). For example, activity of several SP/KLF family members are enriched in SC1, including *Sp1*, which is reported to upregulate the transcription of nectin-2 and JAM-B in Sertoli cells (*Lui and Cheng, 2012*). SC1 is also characterized by a significantly higher gene activity of *Amh* and *Mbd3*, a 5hmC binding protein, when compared with other clusters (*Figure 5E*, *Figure 5—figure supplement 1A*). Immunostaining results confirm the properties of SC1 cells as MBD3-high cells tend to have higher levels of AMH (MBD3-high/AMH-high). It also underscores the heterogeneity of Sertoli cells revealed by scATAC-Seq, as evidenced by the presence of various expression patterns such as MBD3-low/AMH-high (cluster SC3, *Figure 5A*) and MBD3-low/AMH-low (cluster SC2/4/5/6, *Figure 5A*) (*Figure 5F*). We also observed lower gene activity of Sertoli cell markers *Sox9*, *Dmrt1*, and *Wt1* in SC1, as well as the lowest motif binding activity and gene score for FOS, a mediator of Sertoli cell differentiation (*Figure 5E*, *Figure 5—figure supplement 1C*) (*Papadopoulos and Dym, 1994*). Therefore, SC1 cells might represent a less differentiated state of Sertoli cells, which warrant further investigation (*Estermann et al., 2020*).

## Characterization of TF regulation during perinatal Leydig cell development

Re-clustering of Leydig cells generated three main clusters (*Figure 5G*, *Figure 5—figure supplement 1D*). Based on the gene activities of known Leydig cell markers, cluster LC2 is likely to represent FLCs, while LC1 and LC3 showed enrichment in different sets of SLC markers (*Figure 5—figure supplement 1E*). Notably, most of the marker genes identified across the clusters were upregulated in LC1 (*Figure 5—figure supplement 1F*). GO terms associated with the marker genes in LC1 include extracellular structure organization and connective tissue development (*Figure 5—figure supplement 1G*). For instance, *Col4a1* and *Col4a2* showed higher gene activity in LC1, accompanied by LC1-specific peak-to-gene links to the bidirectional collagen IV promoter (*Figure 5H*). Previous studies have indicated that collagen IV-mediated signaling is involved in progenitor Leydig cell proliferation, suggesting that LC1 cells may represent a later developmental stage of SLC (*Anbalagan and Rao, 2004*).

Next, we identified 40 putative positive TF regulators of Leydig cells (*Figure 5I*). For instance, thyroid hormone receptors *Thra* and *Thrb* were upregulated in LC2. Other potential TF candidates include NR and Kruppel-like factor (KLF) family members. These results suggested there is also large heterogeneity among different TFs in their involvement in different Leydig cell subpopulations.

Taken together, we uncovered potential genes and TFs that regulate the three main Leydig cell subpopulations present during the perinatal period.

## Stromal cell heterogeneity and PTM development during the perinatal period

Since PTMs and stromal cells were clustered into a single large cluster (*Figure 1D*) and PTMs are suggested to be derived from interstitial progenitors (*Shen et al., 2020*), PTMs and stromal cells were grouped together for subsequent analysis. Re-clustering of this cell group generated 10 clusters of cells (*Figure 5—figure supplement 2A and B*). Within the clusters, PC8, PC9, and PC10 showed higher gene activity of *Myh11*, indicating their PTM identity, while other clusters were stromal cells (*Figure 5—figure supplement 2C*); 1927 marker genes were identified across the clusters (FDR <0.1) (*Figure 5—figure supplement 2D*). Interestingly, the gene expression pattern of PC2 resembles telocytes, a recently described stromal cell type, which co-expresses *Tcf21*, *Cd34,* and *Pdgfra* and is negative for *Pecam1*, *Kit,* and *Acta2* (*Figure 5—figure supplement 2C*; *Marini et al., 2018*). Our results thus confirm that testis stromal cells are highly heterogeneous in nature.

Motif enrichment analysis of DARs across clusters indicated that AR, PGR (progesterone receptor), and muscle-specific TF motifs were enriched in PTM clusters (*Figure 5—figure supplement 2E*). In contrast, WT1 and SP family members exhibited higher TF motif activity in stromal cell cluster PC3.

We then identified the TFs enriched along the PTM developmental trajectory (*Figure 5—figure supplement 2F and G*). Numerous candidates were associated with steroidogenesis (*Dlx6*, *Egr1*, *Xbp1*, *Sp3*) and spermatogenesis (*Hmga2*, *Nfya*, *Rfx1*, *Tbp*). Interestingly, 9 of the 46 TFs along the trajectory are homeobox genes. For example, *Hoxc6* is upregulated at the early stages and has been implicated in steroid hormone regulation (*Ansari et al., 2011*).

Lastly, we identified the putative positive TF regulators regulating PTM and stromal cells (*Figure 5—figure supplement 2H*). Notably, *Smad3*, *Gata4*, *Tcf15*, *Nhlh2,* and *Ppard* showed upregulation in PTM clusters, while *Rfx1*, *Rfx2*, *Lbx2*, *Nfya*, *Yy1,* and *Fosl1* were upregulated in stromal clusters. Concordantly, *Gata4* has been described as a negative regulator of contractility in PTM, whereas *Smad3* is associated with androgen responsiveness and postnatal testis development (*Itman et al., 2011*; *Wang et al., 2018a*).

## Identification of cell type-specific TFs in immune cells

Re-clustering of all the immune cells generated four-cell clusters (*Figure 1D*, *Figure 5—figure supplement 3A*), which can be re-grouped as three main groups based on the expression of their corresponding marker genes, including T cells/NK cells (IC1 and IC2), myeloid cells (IC3), and dendritic cells (IC4) (*Figure 5—figure supplement 3B*). Since testicular tissue-resident macrophages were shown to be involved in steroidogenesis, spermatogonia differentiation, and Leydig cell function, our subsequent analysis focused on IC3 which included the macrophage population. We first identified cluster-specific positive TF regulators (*Figure 5—figure supplement 3C and D*). This accurately identified C/EBP proteins governing macrophage differentiation and mobilization and KLFs controlling macrophage activation and polarization (*Date et al., 2014*; *Mahabeleshwar et al., 2011*; *McMahon et al., 1989*; *Wada et al., 2015*). Other well-known TFs with functions in myeloid cells/macrophages were identified including *Zeb1*, *Creb1*, *Fosl2*, *Egr1,* and *Nfat5*. Our analysis also revealed new candidates potentially regulating testis myeloid cells, such as *Nfya*, *Zfp42,* and *Tef*. Notably, *Nfya* and *Zfp42* participate in testicular functions (*Iyer et al., 2016*; *Rezende et al., 2011*). TF candidates identified above could serve as future investigation targets related to testicular immune cells.

## Single-cell chromatin accessibility identified human GWAS target regulatory regions, genes, and cell types in the testis

Genome-wide association studies (GWAS) have been exceedingly successful in identifying nucleotide variations associated with specific diseases or traits. The significance of these findings can be realized only when the associated DNA sequence variation is linked to specific genes and the relevant cell types. Therefore, we sought to predict which cell types in the testis may be the functional targets of polymorphisms from previous GWAS as reported previously (*Figure 6A*; *Cusanovich et al., 2018*). Cell type-specific LD score regression using testosterone level GWAS results revealed a significant increase in per-SNP heritability for testosterone level in the Leydig cell peak set (*Figure 6B*). Examining the

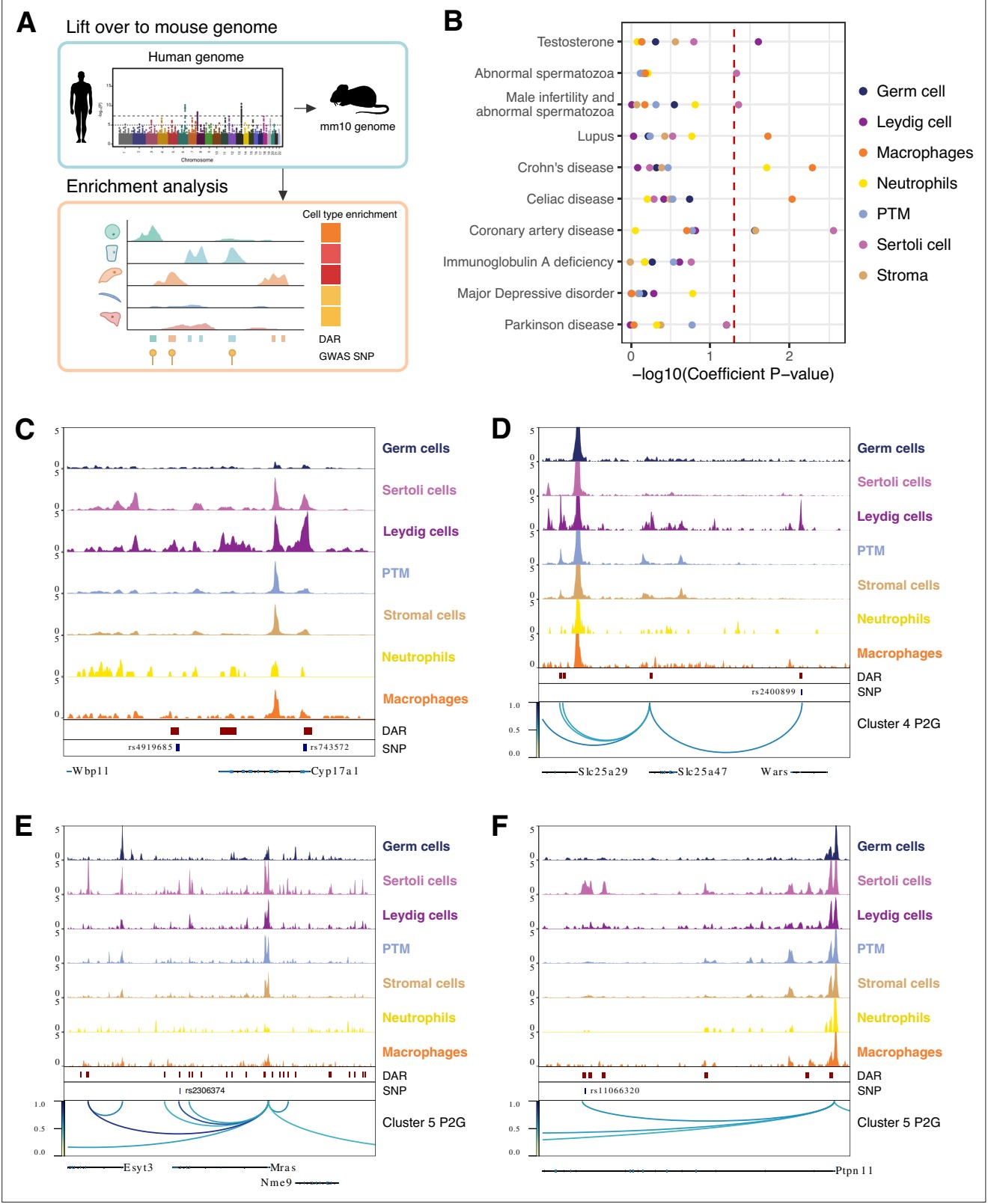

**Figure 6.** Single-cell chromatin accessibility identified human genome-wide association study (GWAS) targets in mouse testis. (**A**) Illustration of heritability enrichment analysis. (**B**) Enrichment of heritability for the selected traits within the cell type-specific differentially accessible chromatin regions (DARs). (**C**) Aggregated single-cell sequencing assay for transposase-accessible chromatin (scATAC-Seq) profiles showing genetic variants (SNPs) overlapped with Leydig cell-specific DARs located at *Cyp17a1* locus. (**D**) Aggregated scATAC-Seq profiles showing SNPs overlapped with Leydig

*Figure 6 continued on next page*

Figure 6 continued

cell-specific DARs and peak-to-gene links to the *Slc25a47* locus. (**E**) Aggregated scATAC-Seq profiles showing SNPs overlapped with Sertoli cell-specific DARs and peak-to-gene links to the *Mras* locus. (**F**) Aggregated scATAC-Seq profiles showing SNPs overlapped with Sertoli cell-specific DARs and peak-to-gene links to the *Ptpn11* locus.

variants overlapping Leydig cell DARs revealed rs4919685 (p=4.203e-15) and rs743572 (p=3.622e-07) located in the TSS-distal regions and the promoter of *Cyp17a1*, respectively (*Figure 6C*). CYP17A1 is one of the enzymes that converts testosterone to estradiol (*Olson et al., 2007*) and rs743572 has been suggested to be a functional SNP related to testosterone levels (*Kakinuma et al., 2004*). Similar analyses in male infertility and abnormal spermatozoa traits showed significant enrichment in SNP heritability in Sertoli cells (*Figure 6B*).

In addition, we predicted immune cell types as the targets of GWAS variants related to several immune diseases, such as lupus, celiac disease, and Crohn's disease. In contrast, the heritability of GWAS SNPs from traits not directly related to testis, such as immunoglobulin A deficiency, Parkinson disease, and major depressive disorder, was not enriched in any of the testicular cell types. Interestingly, we found that the GWAS variants related to coronary artery disease (CAD) were highly enriched in Sertoli cell and stromal cell peaks. This may be linked to the role of androgens in cardiovascular physiology as Sertoli cells are critical regulators of androgen secretion (*Liu et al., 2003*; *Rebourcet et al., 2014*).

We also wanted to predict the genes that may be direct regulatory targets of a given non-coding polymorphism. Using our catalog of cell type-specific peak-to-gene links, we linked TSS-distal GWAS variants to target genes. As an example of the Leydig cell-specific peak-to-gene link and testosterone level, we found the variant within *Wars* (rs2400899: p=4.993e-10) was connected to the nearby gene *Slc25a47*, which encodes a mitochondrial transporter (*Figure 6D*). This might be linked to the mitochondrial functions in regulating Leydig cell steroidogenesis (*Hales et al., 2005*). We then focused on Sertoli cells and CAD traits. We found the variant rs2306374 (p=6.49e-09) was located in a Sertoli cell-specific peak and was linked to *Mras*, which was reported to be associated with CAD (*Figure 6E*) (CARDIoGRAMplusC4D *Deloukas et al., 2013*). We also found rs11066320 (p=1.03e-08) located in the *Ptpn11* locus (*Figure 6F*). The risk allele rs11066320 is robustly associated with HDL cholesterol levels, which are important in determining risk for CAD (*Richardson et al., 2020*).

The data generated here can also be used to help identify the gene targets of non-coding GWAS polymorphisms with a strong association with other traits not assayed above. The genetic variant rs941576 lies within a conserved and regulatory region within intron 6 of the maternally expressed *Meg3* (*Figure 3C*). It has been suggested that rs941576 is strongly associated with T1D susceptibility and rheumatoid arthritis (*Wahba et al., 2020*; *Wallace et al., 2010*). Our data show that this SNP in the mouse genome lies within a peak linked to *Dlk1*. Therefore, it is plausible that this SNP alters the regulation of the paternally expressed *Dlk1*. Taken together, a combination of our scATAC-Seq data with human traits or disease-relevant SNPs enables the prediction of target cell types and target genes for GWAS variants.

## Discussion

Understanding the genetic networks that underlie developmental processes requires a comprehensive understanding of the epigenetic regulatory mechanisms that modulate the expression of key developmental genes. To date, the studies related to the male reproductive system mainly focus on germ cells (*Cheng et al., 2020*; *Lazar-Contes et al., 2020*). There have only been a few studies adopting epigenetic approaches, including DNaseI-Seq and ChIP-Seq, to address the regulation of cell fate in somatic cells in the testis (*Maatouk et al., 2017*; *Maezawa et al., 2021*). These studies also lacked single-cell resolution. Recently, Wu et al. performed scATAC-Seq analysis of adult human testis. Although somatic cell types could be identified based on chromatin accessibility, detailed characterization was not feasible due to the small number of cells captured (*Wu et al., 2022*).

In this study, we present a comprehensive open chromatin map at single-cell resolution for the developing testis. First, we showed that similar to scRNA-Seq, the chromatin accessibility information obtained from scATAC-Seq is able to define cell types in the testis. Second, identification of TFs has been challenging solely based on gene expression information from scRNA-Seq. Our scATAC-Seq

provides additional information on TF regulation by correlating motif accessibility and predicted gene activity, which allows us to reveal cell type-specific TFs. Third, we found numerous peak-to-gene links among different cell types, which demonstrates the tight regulatory association between CREs and gene expression. More importantly, a significant amount of peak-to-gene links are within known testis enhancer regions, indicating the reliability of our analysis strategy. Lastly, our study illustrates the critical role of scATAC-Seq in identifying target cell types of known GWAS variants.

Our study revealed dynamic chromatin accessibility that tracks with germ cell differentiation. Notably, in the pseudotime trajectory analysis of germ cells, besides a conventional pathway from ProSG to differentiating spermatogonia through an undifferentiated spermatogonial state, we also observed a second trajectory in which T1-ProSG reached the differentiating state through two cell clusters with unknown identities. We speculated that this may represent the germ cells undergoing the first wave of spermatogenesis, which are a subset of ProSGs that differentiate immediately into spermatogonia and undergo spermatogenesis in early postnatal life (*Yoshida et al., 2006*). In fact, the low gene expression of *Ngn3* we observe along this trajectory is concordant with the observation that the first round of mouse spermatogenesis initiated directly from ProSGs without passing through the *Ngn3*-expressing undifferentiated spermatogonial stage (*Yoshida et al., 2006*). Since there is currently a lack of marker genes to accurately identify gonocytes which undergo the first wave of spermatogenesis, our scATAC-Seq data uncovered a list of potential markers that warrants further investigation.

Our scATAC-Seq analysis revealed novel cell populations not previously reported in scRNA-Seq studies, such as a distinct subpopulation of Sertoli cells. A recent study investigating chicken sex determination proposed the existence of a unique Sertoli cell subpopulation characterized by lower levels of *Dmrt1* and *Sox9* (*Estermann et al., 2020*). Whether the population we identified shares similar properties with the one described in chickens warrants further investigation.

Cell type-specific chromatin marks, when integrated with GWAS variants, can provide insights into disease causal cell types (*Trynka et al., 2013*). We showed that testicular cell type-specific peaks displayed increased heritability enrichment in cell populations consistent with the known biology. Intriguingly, Sertoli cells show higher heritability enrichment for CAD phenotypes, which leads us to hypothesize that CAD-associated variants may act in Sertoli cells. The risk of CAD has been linked to testosterone level, increasing age and male gender (*Nettleship et al., 2009*). Our hypothesis can be supported by the role of Sertoli cells in testosterone level regulation through influencing the Leydig cell function and testicular vasculature (*Rebourcet et al., 2017*; *Rebourcet et al., 2016*). Furthermore, co-accessibility measurement informs well-studied CAD-relevant genes, such as *Mras* and *Ptpn11* in Sertoli cells. These results define immediately testable hypotheses in which these variants modulate the activity of the CREs in Sertoli cells in the context of CAD. While our analysis provided new insights toward the target cell types of a certain SNP, it should be noted that our data were generated on mouse testis. Whether a similar implication could be translated into human testis needs to be validated. Several studies have used scRNA-Seq analysis to characterize human testicular cell types, including somatic cells (*Guo et al., 2018*; *Hermann et al., 2018*; *Sohni et al., 2019*; *Wang et al., 2018b*), which provided significant insights into the testis development. We envision that when our analysis framework is applied to humans and other species, there may be rich opportunities for investigating the evolution of cell type-specific *cis*-acting regulatory elements in male reproductive system.

In summary, our high-resolution data has enabled detailed reconstruction of the gene regulatory landscape of testicular cell populations during key time points in testis development. Functional insight is further revealed by integrating multiple sources of genomic information and when combined provides an invaluable and unique resource for further investigation of key developmental events in the testis.

## Limitations of the study

We acknowledge an important limitation of our study is that we correlated scATAC-Seq profiles with scRNA-Seq data obtained from published dataset. Future studies that incorporate multiomics data will be important for obtaining a more comprehensive understanding of the cellular processes we have studied. Another limitation of our study is that we did not perform experimental assignment of regulatory sequences with gene promoters. Investigating the 3D organization of chromatin and the interactions between regulatory elements and gene promoters using techniques such as Hi-C

has the potential to provide valuable insights and overcome this limitation. Finally, our analysis did not directly measure protein level, and therefore, may not fully capture the functional status of a cell. Complementary techniques such as mass spectrometry or immunofluorescence to validate or confirm protein expression would certainly make it more comprehensive. The integration of single-cell protein measurements may provide even deeper insights into cellular function.

We believe that these limitations provide opportunities for future research to build upon the findings of our study and to further enhance our understanding of the complex regulatory networks that govern testis development.

# Materials and methods

## Animals

All the animal experiments were performed according to the protocols approved by the Animal Experiment Ethics Committee (AEEC) of The Chinese University of Hong Kong (CUHK) and followed the Animals (Control of Experiments) Ordinance (Cap. 340) licensed from the Department of Health, the Government of Hong Kong Special Administrative Region. All the mice were housed under a cycle of 12 hr light/dark and kept in ad libitum feeding and controlled the temperature of 22–24°C. Oct4-EGFP transgenic mice (B6; CBA-Tg(Pou5f1-EGFP)2Mnn/J, Stock no.: 004654) were acquired from The Jackson Laboratory (*Ohbo et al., 2003*). Oct4-EGFP transgenic mice and C57BL/6J mice were maintained in CUHK Laboratory Animal Services Centre.

## Sample collection

The testes of C57BL/6J mice at E18.5, P0, P2, and P5 were collected and with tunica albuginea removed. Each sequencing sample was from three independent mice mixed in equal proportion according to the same cell count to minimize the effects of biological variability. The testes were then digested with 1 mg/ml type 4 collagenase (Gibco), 1 mg/ml hyaluronidase (Sigma-Aldrich), and 5 µg/ml DNase I (Sigma-Aldrich) at 37°C for 20 min with occasional shaking. The suspension was passed through a 40 µm strainer cap (BD Falcon) to yield a uniform single-cell suspension.

## Fluorescence-activated cell sorting

The testes of C57BL/6J mice were collected and with tunica albuginea removed. Isolated seminiferous tubules were digested with 1 mg/ml type 4 collagenase (Gibco), 1 mg/ml hyaluronidase (Sigma-Aldrich) and 5 µg/ml DNase I (Sigma-Aldrich) at 37°C for 20 min with occasional shaking. The suspension was passed through a 40 µm strainer cap (BD Falcon) to yield a uniform single-cell suspension. After incubation in staining buffer (PBS supplemented with 1% FBS, HEPES, glucose, pyruvate, and penicillin-streptomycin) with mouse DLK1 antibody (MAB8634, R&D Systems) at 4°C for 30 min, followed by secondary antibody at 4°C for 30 min. DLK1+ cells were collected with a BD FACSAria Fusion Flow Cytometer (BD Biosciences). Data were analyzed by FlowJo software (FlowJo, LLC).

## scATAC-Seq library preparation and data analysis

### Cell lysis and tagmentation

To minimize technical bias, all sequencing samples were processed in one batch. Cell tagmentation was performed according to SureCell ATAC-Seq Library Prep Kit (17004620, Bio-Rad) User Guide (10000106678, Bio-Rad) and the protocol based on Omni-ATAC was followed (*Corces et al., 2017*). In brief, washed and pelleted cells were lysed with the Omni-ATAC lysis buffer containing 0.1% NP-40, 0.1% Tween-20, 0.01% digitonin, 10 mM NaCl, 3 mM $MgCl_2$, and 10 mM Tris-HCl pH 7.4 for 3 min on ice. The lysis buffer was diluted with ATAC-Tween buffer that contains 0.1% Tween-20 as a detergent. Nuclei were counted and examined under microscope to ensure successful isolation. Same number of nuclei were subjected to tagmentation with equal ratio of cells/Tn5 transposase to minimize potential batch effect. Nuclei were resuspended in tagmentation mix, buffered with 1× PBS supplemented with 0.1% BSA and agitated on a ThermoMixer for 30 min at 37°C. Tagmented nuclei were kept on ice before encapsulation.

## Library preparation and sequencing

Tagmented nuclei were loaded onto a ddSEQ Single-Cell Isolator (Bio-Rad). scATAC-Seq libraries were prepared using the SureCell ATAC-Seq Library Prep Kit (17004620, Bio-Rad) and SureCell ATAC-Seq Index Kit (12009360, Bio-Rad). Bead barcoding and sample indexing were performed with PCR amplification as follows: 37°C for 30 min, 85°C for 10 min, 72°C for 5 min, 98°C for 30 s, eight cycles of 98°C for 10 s, 55°C for 30 s and 72°C for 60 s, and a single 72°C extension for 5 min to finish. Emulsions were broken and products were cleaned up using Ampure XP beads. Barcoded amplicons were further amplified for eight cycles. PCR products were purified using Ampure XP beads and quantified on an Agilent Bioanalyzer (G2939BA, Agilent) using the High-Sensitivity DNA kit (5067-4626, Agilent). Libraries were sequenced on HiSeq 2000 with 150 bp paired-end reads.

## Sequencing reads preprocessing

Sequencing data were processed using the Bio-Rad ATAC-Seq Analysis Toolkit. This toolkit is a streamlined computational pipeline, including tools for FASTQ debarcoding, read trimming, alignment, bead filtration, bead deconvolution, cell filtration, and peak calling. The reference index was built upon the mouse genome mm10. For generation of the fragments file, which contain the start and end genomic coordinates of all aligned sequenced fragments, sorted bam files were further processed with 'bap-frag' module of BAP (https://github.com/caleblareau/bap, v0.6.0) (*Lareau et al., 2019*) . Downstream analysis was performed in ArchR (*Granja et al., 2020*).

## Dimensionality reduction, clustering, and gene score/TF activity analysis

Fragment files were used to create the Arrow files in the ArchR package. We filtered out low-quality nuclei with stringent selection criteria, including read depth per cell (>2000) and TSS enrichment score (>4). Potential doubles were further removed based on the ArchR method. Bin regions were cleaned by eliminating bins overlapping with ENCODE Blacklist regions, mitochondrial DNA, as well as the top 5% of invariant features (house-keeping gene promoters). Dimensionality reduction and clustering was performed with ArchR. Briefly, we used LSI dimensionality reduction using a TFIDF normalization function, UMAP low-dimensional embedding, and clustering using a nearest neighbor graph performed on data in LSI space. To facilitate major cell type annotation, we used harmony to integrate datasets of four time points and project cells onto a shared embedding in which cells were grouped by cell type rather than developmental stage (*Korsunsky et al., 2019*). The clustering analysis within each cell type was carried out without harmony integration.

ArchR was used to estimate gene expression for genes and TF motif activity from single-cell chromatin accessibility data. Gene scores were calculated using the addGeneScoreMatrix function with gene score models implemented in ArchR. addDeviationsMatrix function was used to compute enrichment of TF activity on a per-cell basis across all motif annotations based on chromVAR (chromVAR_0.3). The scATAC data was integrated with scRNA-Seq data (GEO: GSE130593) using the ArchR function addGeneIntegrationMatrix. The integration process involved directly aligning cells from scATAC-Seq with cells from scRNA-Seq using the FindTransferAnchors function from the Seurat package to identify the integrated anchors with default parameters (reduction = 'cca', nGenes = 2000). Marker gene activity scores were used for initial annotation while labels from the RNA integration were used to validate and refine the initial annotations to produce the final cell-type annotations.

## Pseudo-bulk coverage analysis

To create the 'pseudo-bulk' coverage profile for each individual cell type, we split the aligned BAM files of each developmental time point into given groups of cell barcodes belonging to different cell types using sinto tools (https://timoast.github.io/sinto/). deepTools suite was used to convert individual bam files into standard coverage tracks (bigWig files) with RPKM normalization and visualized with pyGenomeTracks (*Ramírez et al., 2018*).

## Trajectory analysis

Trajectory analysis was performed in ArchR. addTrajectory function in ArchR was used to construct trajectory on cisTopic UMAP embedding. To perform integrative analyses for identification of positive TF regulators by integration of gene scores with motif accessibility across pseudotime, we used

the correlateTrajectories function which takes two SummarizedExperiment objects retrieved from the getTrajectories function.

## Footprinting analysis

Differential TF footprints across cell types were identified using the Regulatory Genomics Toolbox application HINT (*Li et al., 2018*). Aligned BAM files from different cell types were treated as pseudo-bulk ATAC-Seq profiles and then subjected to rgt-hint analysis. Based on MACS calling peaks, we used HINT-ATAC to predict footprints with the 'rgt-hint footprinting' command. We then identified all binding sites of a particular TF overlapping with footprints by using its motif from JASPAR with 'rgt-motifanalysis matching' command. Differential motif occupancy was identified with 'rgt-hint differential' command and '–bc' was specified to use the bias-corrected signal.

## Real-time PCR

Total RNA was extracted from cells by AllPrep DNA/RNA Micro Kit (QIAGEN). RNA was then converted to cDNA by reverse transcription using PrimeScript RT Master Mix (Takara). Real-time PCR was performed using Power SYBR Green PCR Master Mix (Life Technology) following the manufacturer's instructions. Primers used: *Igf1* forward (5'–3') GGACCGAGGGGCTTTTACTT, reverse (5'–3') GTGGGGCACAGTACATCTCC; *Cyp11a1* forward (5'–3') CACAGACGCATCAAGCAGCAAAA, reverse (5'–3') GCATTGATGAACCGCTGGGC; *Acta2* forward (5'–3') GCTGGTGATGATGCTCCCA, reverse (5'–3') GCCCATTCCAACCATTACTCC.

## Immunofluorescence staining of testis sections

Testis sections on microscope slides were fixed in 4% (vol/vol) paraformaldehyde for 15 min at room temperature and rinsed three times in PBS for 5 min before staining. Cells were permeabilized by treating in PBS containing 0.2% Triton X-100 for 15 min. Cells were blocked for 2 hr in PBS containing 10% normal donkey serum (Jackson Immunoresearch). Primary and secondary antibodies and respective dilutions were listed in *Supplementary file 6*. Primary antibodies were applied overnight at 4°C. Appropriate secondary antibodies were applied for 1 hr at room temperature. Cell nuclei were counterstained with DAPI (1 μg/ml, Sigma). All confocal images were captured by Olympus FV1200 confocal microscope. FV10-ASW software (Olympus) was employed to create overlays of colors.

## Statistical analysis

Assessment of statistical significance was performed using two-tailed unpaired t-tests, one-way ANOVA with Tukey multiple comparisons tests or Chi-squared tests. Statistical analysis was performed using GraphPad Prism v8. Associated p-values are indicated as follows: *$p<0.05$; **$p<0.01$; ***$p<0.001$; ****$p<0.0001$; not significant (ns) $p>0.05$.

## Acknowledgements

This study was supported by the Department of Chemical Pathology's Faculty Startup Fund (CUHK) to JL and the General Research Fund (CUHK 14120418) to TLL. We would like to acknowledge the technical support provided by the SBS Core Laboratories and express our gratitude to the Biomedical Computing Centre (BCC) at the Li Ka Shing Institute of Health Sciences (LiHS) in The Chinese University of Hong Kong for offering computing power and data storage.

## Additional information

### Funding

| Funder | Grant reference number | Author |
| --- | --- | --- |
| Chinese University of Hong Kong | Department of Chemical Pathology's Faculty Startup Fund | Jinyue Liao |

| Funder | Grant reference number | Author |
|---|---|---|
| University Grants Committee | General Research Fund CUHK 14120418 | Tin-lap Lee |

The funders had no role in study design, data collection and interpretation, or the decision to submit the work for publication.

## Author contributions
Hoi Ching Suen, Conceptualization, Data curation, Formal analysis, Validation, Investigation, Visualization, Methodology, Writing – original draft, Writing – review and editing; Shitao Rao, Hon Cheong So, Software, Formal analysis, Methodology; Alfred Chun Shui Luk, Validation, Methodology, Writing – review and editing; Ruoyu Zhang, Software, Formal analysis; Lele Yang, Huayu Qi, Validation; Robin M Hobbs, Investigation, Writing – review and editing; Tin-lap Lee, Resources, Supervision, Funding acquisition; Jinyue Liao, Conceptualization, Resources, Data curation, Software, Formal analysis, Supervision, Funding acquisition, Validation, Investigation, Visualization, Methodology, Writing – original draft, Project administration, Writing – review and editing

## Author ORCIDs
Hoi Ching Suen ⬚ http://orcid.org/0000-0003-4060-8603
Hon Cheong So ⬚ http://orcid.org/0000-0002-7102-833X
Tin-lap Lee ⬚ http://orcid.org/0000-0002-6654-0988
Jinyue Liao ⬚ http://orcid.org/0000-0003-3392-3760

## Ethics
All the animal experiments were performed according to the protocols approved by the Animal Experiment Ethics Committee (AEEC) of The Chinese University of Hong Kong (CUHK) and followed the Animals (Control of Experiments) Ordinance (Cap. 340) licensed from the Department of Health, the Government of Hong Kong Special Administrative Region.

## Decision letter and Author response
Decision letter https://doi.org/10.7554/eLife.75624.sa1
Author response https://doi.org/10.7554/eLife.75624.sa2

# Additional files

## Supplementary files
• Supplementary file 1. List of top 2000 variable genes identified from scRNA-Seq data used in the integration analysis.
• Supplementary file 2. List of all peaks (accessible chromatin regions) identified from all cells.
• Supplementary file 3. Sequencing metrics of single-cell sequencing assay for transposase-accessible chromatin (scATAC-Seq) libraries.
• Supplementary file 4. List of differentially chromatin accessible regions (DARs) of each cell type.
• Supplementary file 5. List of peak-to-gene links identified across cell types.
• Supplementary file 6. List of antibodies and dilutions.
• MDAR checklist

## Data availability
All raw and processed sequencing data generated in this study have been submitted to the NCBI Gene Expression Omnibus (GEO; https://www.ncbi.nlm.nih.gov/geo/) under accession number GSE164439. Code for producing the majority of analyses from this paper is available on GitHub at https://github.com/liaojinyue/mouse_testis_scATAC, (copy archived at *Liao, 2023*).

The following dataset was generated:

| Author(s) | Year | Dataset title | Dataset URL | Database and Identifier |
|---|---|---|---|---|
| Liao J, Suen HC, Lee T | 2021 | Genome-wide maps of chromatin state in mouse perinatal testes [scATAC-seq] | https://www.ncbi.nlm.nih.gov/geo/query/acc.cgi?&acc=GSE164439 | NCBI Gene Expression Omnibus, GSE164439 |

The following previously published dataset was used:

| Author(s) | Year | Dataset title | Dataset URL | Database and Identifier |
|---|---|---|---|---|
| Tan K, Song H, Wilkinson MF | 2020 | Single-cell RNAseq analysis of testicular germ and somatic cell development during the perinatal period | https://www.ncbi.nlm.nih.gov/geo/query/acc.cgi?acc=GSE130593 | NCBI Gene Expression Omnibus, GSE130593 |

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
