## [Editor Report]

This manuscript by Liao et al. aims to understand the genetic networks that underlie or modulate gonadogenesis and germ cell maturation during the fetal to neonatal transition. This goal was achieved by performing scATACseq on multiple timepoints (E18.5 and Postnatal days 1,2,5). Clustering of thousands of cells has striking cellular diversity and convincingly led to the identification of additional novel populations, of both germ cell and somatic origins. This is an important paper with far-reaching implications in reproductive biology, but additional validation would be needed to confirm the correlative observations and the functionality of newly identified testis cells.

---

## [Decision Letter]

**Decision letter after peer review:**

Thank you for submitting your article "The single-cell chromatin accessibility landscape in mammalian perinatal testis development" for consideration by *eLife*. Your article has been reviewed by 2 peer reviewers, and the evaluation has been overseen by a Reviewing Editor and Marianne Bronner as the Senior Editor. The reviewers have opted to remain anonymous.

Essential revisions:

Overall, the two external experts agreed that this study may offer new and exciting insights into the cell populations that support spermatogenesis at birth. However, because the scATAC-seq was performed on a single replicate, there is no statistical power to support the strength of the main findings, i.e. the identification of novel cell types. Therefore, there is a important need for independent validation.

Please refer to the two points expanded below to prepare your revisions, and please consult the detailed points raised by the reviewers, for further corrections or analytical additions, rephrasing of the manuscript and support for discussion in a point-by-point rebuttal.

1) The quality of immunofluorescence data is not optimal and the validation results often often inconclusive. Better independent validation is needed to support the identification of novel populations by scATAC-seq, a point that was raised by the two reviewers. The results that NR5A1 may mark germ cells that commit to spermatogenesis in an SSC-independent manner are intriguing. However, whole-mount immunostaining provided in Figure 4E is not convincing. Clear colocalization between NR5A1 and germ cell markers should be provided, by cell cytometry and/or IF on sections. Clearer validation that Sertoli stem cells were identified is also needed, the whole-mount staining in Figure 5F being poorly informative. Reviewer #1 provides useful suggestions to make progress on that manner. Reviewer #2 also requests additional analysis to support this identification.

2) One limitation of the study is to correlate scATAC-seq profiles generated here with scRNA-seq data available in the literature and obtained on cell populations that may not be defined according to the same experimental and analytical tools. It is not requested here to perform multiomics on the same cells, but this limitation should be acknowledged. The other limitation comes from using correlation between the presence of TF binding sites at open chromatin sites and gene expression of nearby genes, without proper experimental assignment of regulatory sequences with gene promoters by 3D assays. This experiment would be very difficult to carry on in this developmental and cellular context. Nonetheless, this limitation should also be acknowledged.

3) Replace "mammalian" with "mouse" in the title.

*Reviewer #1 (Recommendations for the authors):*

In supplemental figure 2 – projecting the dividual datasets will help the reader evaluate the contribution of the individual datasets to each cluster.

A summary stat's table about individual cell quality metrics would be helpful. please detail inclusion / exclusion criteria # peaks per cell etc…

The alignment of scRNAseq clusters with scATACseq was missing in S2D.

What is a deviation score in Figure S3B? Is this the same as a correlation?

EMT in germ cells has also been previously described in http://genesdev.cshlp.org/content/29/21/2312

Based on the text and figures for supplemental figure 11 it is hard to fully appreciate the stromal/myoid cell heterogeneity. The authors identify 10 clusters – PC2 appears to be called out as a TCf21 and telocyte population – is this correct? Are these the same the same population but classified differently?

This paper by Liao et al. aims to understand the genetic networks that underlie or modulate gonadogenesis and germ cell maturation during the fetal to neonatal transition. This goal was achieved by performing scATACseq on multiple timepoints (E18.5 and Postnatal days 1,2,5). Clustering of thousands of cells has revealed striking cellular diversity and led to the identification of additional novel populations. This is an exciting paper that may have far reaching implications, but additional validation is needed for novel populations identified.

*Reviewer #2 (Recommendations for the authors):*

1) It appears that for each time point, only one sample was analyzed, meaning that there was no statistical power to discern whether the outcome is a random observation or true reflection of a biological event. If having statistically meaning sample size is impossible, a duplication of samples should have been done at least to exclude potential technical deviation. In addition, it is not clear whether for each sample, it is a pool of several testes or a single testis.

2) One concern is the lack of details and reference gene lists for making correlations between the published single cell RNAseq dataset from Tan et al. and the single cell ATACseq dataset from this study. This is the critical first step to validate the scATACseq results and assign cell identities. Only one gene per cell type was given as an example.

3) Sertoli and Leydig cells are distinct cell types in terms of their functions and transcriptional profiles. However, the scATACseq dataset did not detect cell specific TF binding profile (Figure 2C). This is quite puzzling.

4) The presence of TF binding sites in the open chromatin region is a suggestion, not a confirmation of TF activity. The authors are advised to make clarification on this matter and tone down the language on such correlation without further validation like ChIP-seq. For example, in Figure 2H, *SOX9* binding motifs are enriched in both Sertoli and germ cell cluster but its expression is absent in the germ cells. Similarly, TCF21 motifs and gene expression are enriched in a specific cluster of stromal cells but TCF21 gene activity is not different from other cell type. It is not clear how authors reconcile these conflict findings.

5) Making correlation between separated scATACseq data and scRNAseq data has its limitation. One still cannot be certain that the cell from scATACseq is the same as the one from scRNAseq. This is the major limitation of this study.

6) Similarly, the chromatin interaction (Figure 3A-F) is based on correlation between open chromatin and gene expression. Without confirmation of 3C and High C analyses, the results are strictly speculative.

7) Some of the figures are difficult to follow with insufficient description of how the experiments were conducted. For example, experiments for Figure 3F were not described in the result session.

8) The findings of NR5a1 as a potential regulator of an unknown population of male germ cells are interesting but puzzling. NR5a1 has never been shown to be expressed in the testis by cells other than the somatic cell populations. The wholemount colocalization image is far from convincing.

9) Similarly, the unique expression of MBD3 in a subset of Sertoli cells is quite fascinating. However, the analysis was superficial and inconclusive. This subset of Sertoli cells should be *SOX9* negative/low and AMH positive/high. Unfortunately, such confirmation is not available to support the conclusion.

10) The intent to link human GWAS data with the chromatin status of various cell types is reasonable and novel. Unfortunately, confirmation of the cell type-specific expression of several putative target genes (Wars, Mras, Ptpn11, etc) was not provided. The same problem is found in other analyses on Sertoli and Leydig cell stem cell types.

---

## [Author Response]

Reviewer #1 (Recommendations for the authors):In supplemental figure 2 – projecting the dividual datasets will help the reader evaluate the contribution of the individual datasets to each cluster.

We agree that this would provide more clarity for the reader and we have included this in our revised manuscript (Figure 1—figure supplement 2).

A summary stat's table about individual cell quality metrics would be helpful. please detail inclusion / exclusion criteria # peaks per cell etc…

We have summarized individual cell quality metrics in Supplementary File 3. The details regarding our inclusion/exclusion criteria for cells has been mentioned in the Materials and methods section.

The alignment of scRNAseq clusters with scATACseq was missing in S2D.

Thank you for pointing out this mistake. We apologize for the confusion and have corrected the labeling in the revised manuscript. The correct figure that shows the prediction score aligning scRNAseq clusters with scATACseq is Figure 1—figure supplement 2C.

What is a deviation score in Figure S3B? Is this the same as a correlation?

We apologize for any confusion caused by our unclear explanation of the figure. In Figure 2—figure supplement 1B (Figure S3B in original version), we used ChromVAR to calculate the deviation score of Sertoli cell DNase I hypersensitive sites (DHSs) in our scATAC-seq dataset and observed a higher deviation score in the Sertoli cluster. This suggests that the accessibility of the Sertoli cell DHSs in this cluster deviates significantly from the average accessibility of these sites in all other clusters. To further confirm the identity of this cluster as Sertoli cells, we have added feature enrichment analysis in the revised manuscript. We showed that the cluster-specific ATAC-seq peaks for the Sertoli cell cluster are significantly enriched for Sertoli cell DHSs compared to other clusters. This supports our finding that this cluster represents Sertoli cells with unique chromatin accessibility profiles. We hope that this explanation clarifies how we confirmed Sertoli cell clusters in our scATAC-seq data.

EMT in germ cells has also been previously described in http://genesdev.cshlp.org/content/29/21/2312

We appreciate the reviewer bringing this paper to our attention. We are aware of this study and it provides interesting insights into the heterogeneity of germ cells and their plasticity during development, which is aligned with our finding. We have referenced this paper in our revised manuscript. In addition, we have conducted additional studies and our results are currently being prepared for publication in a follow-up manuscript

(https://www.biorxiv.org/content/10.1101/2020.10.12.336834v2).

Based on the text and figures for supplemental figure 11 it is hard to fully appreciate the stromal/myoid cell heterogeneity. The authors identify 10 clusters – PC2 appears to be called out as a TCf21 and telocyte population – is this correct? Are these the same the same population but classified differently?

In Figure 5 – —figure supplement 2 (supplemental figure 11 in original version), our rationale for suggesting the possible identity of PC2 cluster as telocytes is based on a recent publication by Marini et al. (2018) that reports telocytes as a stromal cell type present in various tissues, including the testis (PMID:30283023). In line with this paper, we found that the PC2 cluster displayed higher activity of markers for telocytes such as Tcf21, Cd34 and Pdgfra. However, we acknowledge that further validation and functional studies are needed to confirm this hypothesis. In the revised manuscript, we wanted to avoid any confusion and have deleted the introduction of Tcf21’s function in bipotent somatic cells because it’s less relevant to the postnatal testis. We hope we have made it more clear.

Reviewer #2 (Recommendations for the authors):1) It appears that for each time point, only one sample was analyzed, meaning that there was no statistical power to discern whether the outcome is a random observation or true reflection of a biological event. If having statistically meaning sample size is impossible, a duplication of samples should have been done at least to exclude potential technical deviation. In addition, it is not clear whether for each sample, it is a pool of several testes or a single testis.

We appreciate the reviewer's concern regarding the statistical power of our study. Due to resource constraints, we were only able to analyze one sequencing sample per time point. Each sequencing sample was from 3 independent mice mixed in equal proportion according to the same cell count to minimize the effects of biological variability. Also, all the sequencing samples were processed in a single cell capture run, which should minimize the technical bias. We believe that our stringent quality control measures and experimental approach have minimized the effects of technical variability. We have carefully mentioned the details in the Materials and methods section.

2) One concern is the lack of details and reference gene lists for making correlations between the published single cell RNAseq dataset from Tan et al. and the single cell ATACseq dataset from this study. This is the critical first step to validate the scATACseq results and assign cell identities. Only one gene per cell type was given as an example.

We have added more details of integration analysis in the Materials and methods section. Gene used in the prediction are the top 2000 variable genes from scRNA-seq reference dataset (Supplementary File 1). The integration process involved directly aligning cells from scATAC-seq with cells from scRNA-seq using the FindTransferAnchors() function from the Seurat package to identify the integrated anchors with default parameters (reduction = “cca”, nGenes = 2000). It is worth noting that the top variable genes are typically cell type-specific genes that exhibit high expression variability across cells and are therefore highly informative for cell type classification and identity assignment.

3) Sertoli and Leydig cells are distinct cell types in terms of their functions and transcriptional profiles. However, the scATACseq dataset did not detect cell specific TF binding profile (Figure 2C). This is quite puzzling.

We appreciate the reviewer's comment regarding the lack of cell-specific TF binding profiles between Sertoli and Leydig cells in Figure 2C. We have investigated this issue and found that the criteria to include TF in the heatmap were stringent. Therefore, we have re-analyzed the data and updated the heatmap to show the differential enrichment of TF motifs in Sertoli and Leydig cells. We found that ESR2 motif is more enriched in Leydig cells, suggesting that it could be a cell-specific TF in this cell type (PMID: 15823796). In addition, the updated results show that Leydig cells exhibit a higher enrichment of several TF motifs such as Nhlh2 and Ascl1/2 compared to Sertoli cells. Enrichment of Nhlh2 and Ascl1/2 in Leydig cells is shared with peritubular myoid (PTM) cells, likely because they share the same cellular origin during gonad development (PMID:30893600). We have updated Figure 2C to reflect these findings. Thank you for bringing this to our attention.

4) The presence of TF binding sites in the open chromatin region is a suggestion, not a confirmation of TF activity. The authors are advised to make clarification on this matter and tone down the language on such correlation without further validation like ChIP-seq. For example, in Figure 2H, SOX9 binding motifs are enriched in both Sertoli and germ cell cluster but its expression is absent in the germ cells. Similarly, TCF21 motifs and gene expression are enriched in a specific cluster of stromal cells but TCF21 gene activity is not different from other cell type. It is not clear how authors reconcile these conflict findings.

We appreciate the reviewer's comment and we acknowledge that the presence of TF binding sites in the open chromatin region is not a confirmation of TF activity. We understand the importance of further validation, such as ChIP-seq, to confirm TF activity. As a suggestion rather than a confirmation, we believe that the results we provide can still offer insights and candidate genes for follow-up studies. Due to the limitation of TF activity prediction, it is challenging to identify the specific TFs that drive observed changes in chromatin accessibility, especially when TFs from the same family share similar motifs. *Sox9* showed its enrichment of TF activity in both Sertoli and germ cell clusters, despite its known expression only in Sertoli cells. This could be due to other SOX family genes, such as SOX3, which have similar DNA binding motifs (PMID:15893302). This is why we believe that our positive regulator analysis approach can overcome this limitation to some extent and provide more informative results. In this case, *SOX9* shows concordant enrichment of TF activity and gene expression in sertoli cells and meets this criteria of a positive regulator and should be regarded as a good candidate in Sertoli cells but not in germ cells. We understand that this approach may not be immediately clear to the reader and have further clarified this in our manuscript.

To improve clarity in Figure 2H, we will only show TF activity and gene expression and clarify that the positive regulator score is calculated based on the correlation between these two measurements. The readers can refer to our website for more information on gene activity.

Finally, we have added some discussion in the manuscript to address these concerns and to emphasize the limitations and potential pitfalls of using TF binding motifs as suggestions for TF activity. We hope this will help readers approach the data more critically and come up with their own candidate genes.

5) Making correlation between separated scATACseq data and scRNAseq data has its limitation. One still cannot be certain that the cell from scATACseq is the same as the one from scRNAseq. This is the major limitation of this study.

The reviewer raises a valid concern regarding the limitations of making correlations between separated scATAC-seq and scRNA-seq datasets.While we cannot definitively confirm that each cell in the scATAC-seq and scRNA-seq datasets are exact matches, we took steps to minimize the potential impact of this issue. Specifically, we carefully matched the time points between the scATAC-seq and scRNA-seq datasets, which allowed us to better infer the transcriptional activity of each cell type during the developmental stages of the testis. Additionally, to ensure a comprehensive representation of cell types and avoid potential biases, we opted not to use pre-selection techniques such as FACS sorting. We believe that our data and analysis provide a robust foundation for further investigations into the transcriptional regulation of testicular cell development.

6) Similarly, the chromatin interaction (Figure 3A-F) is based on correlation between open chromatin and gene expression. Without confirmation of 3C and High C analyses, the results are strictly speculative.

We acknowledge the limitations of our study with regard to the lack of experimental validation of chromatin interactions. While our approach for inferring interactions based on correlation between open chromatin and gene expression is a widely used and effective method (for example in PMID: 33850129), we recognize that this correlation does not necessarily indicate a direct physical interaction. Future studies, such as those employing techniques like Hi-C, could provide further insights into the 3D organization of chromatin and the interactions between regulatory elements and gene promoters. We believe that these limitations highlight opportunities for further research to build upon our findings and to enhance our understanding of the complex regulatory networks that govern cellular processes.

7) Some of the figures are difficult to follow with insufficient description of how the experiments were conducted. For example, experiments for Figure 3F were not described in the result session.

We apologize for any confusion that may have arisen due to the insufficient description of experiments in our manuscript. We have now revised the manuscript to provide a more comprehensive and detailed account of the experiments and methods employed, specifically for Figure 3F. Our goal is to ensure the transparency and reproducibility of our research, and we have taken care to clearly describe the methods and experimental details in the revised manuscript, both in the main text and the Materials and methods section. If further clarification is needed, we are more than willing to provide additional information upon request.

8) The findings of NR5a1 as a potential regulator of an unknown population of male germ cells are interesting but puzzling. NR5a1 has never been shown to be expressed in the testis by cells other than the somatic cell populations. The wholemount colocalization image is far from convincing.

We have performed immunostaining of NR5A1 in testicular sections and showed that NR5A1+ germ cells (TRA98+ cells) exist in P5.5 testis. As additional evidence supporting our finding that a subset of somatic markers are expressed in the unique germ cell population we identified, we reference a study where cells in the spermatogonial signature 3 cluster showed high levels of mRNAs characteristic of Sertoli cells, including Nr5a1, *Sox9*, and Wt1 (PMID: 25568304). This indicates that cells with germ cell identity can express somatic cell genes, which is consistent with our findings. Additionally, another study reported the expression of the somatic cell marker WT1 in some germ cells through immunostaining (Figure 3B, PMID: 34815802). We have included this information in the revised manuscript to further support our conclusion.

9) Similarly, the unique expression of MBD3 in a subset of Sertoli cells is quite fascinating. However, the analysis was superficial and inconclusive. This subset of Sertoli cells should be SOX9 negative/low and AMH positive/high. Unfortunately, such confirmation is not available to support the conclusion.

Following the reviewer’s suggestions, we conducted further immunostaining of MBD3 and AMH in Sertoli cells (Figure 5F), which showed that MBD3-high cells tend to have higher levels of AMH expression. On the other hand, the antibodies that worked well in our hands for both MBD3 and *SOX9* were raised in rabbit species, making it infeasible to perform co-staining due to the overlap in the secondary antibodies. We made efforts to find suitable *SOX9* antibodies raised in other species, but unfortunately, none of them provided satisfactory staining results. This limitation hindered our ability to perform a direct co-staining of MBD3 and *Sox9*.

Despite this challenge, we have provided additional data on MBD3 and AMH staining to offer a more comprehensive understanding of the MBD3-expressing Sertoli cell subpopulation. The observed staining results not only confirm the properties of MBD3+ cells (MBD3-high/AMH-high) but also highlight the heterogeneity of Sertoli cells, as evidenced by the presence of various expression patterns such as MBD3-low/AMH-high (cluster SC3 in Figure 5A) and MBD3-low/AMH-low (cluster SC2/4/5/6 in Figure 5A). We hope that this explanation clarifies the difficulties we encountered and that our revised data adequately addresses your concerns.

10) The intent to link human GWAS data with the chromatin status of various cell types is reasonable and novel. Unfortunately, confirmation of the cell type-specific expression of several putative target genes (Wars, Mras, Ptpn11, etc) was not provided. The same problem is found in other analyses on Sertoli and Leydig cell stem cell types.

We acknowledge the importance of confirming the cell type-specific expression of putative target genes identified from our analysis and have mentioned this limitation in the Discussion section. As a starting point, our study has provided a comprehensive list of putative target genes and their associated GWAS trait in our manuscript, which we hope will facilitate further studies aimed at confirming their expression and function in relevant cell types.